# A COVID-19 peptide vaccine for the induction of SARS-CoV-2 T cell immunity

Jonas S. Heitmann[1,2,14], Tatjana Bilich[1,2,3,14], Claudia Tandler[1,2,3,14], Annika Nelde[1,2,3], Yacine Maringer[1,2,3], Maddalena Marconato[1], Julia Reusch[1], Simon Jäger[1,4,5], Monika Denk[3], Marion Richter[3], Leonard Anton[1], Lisa Marie Weber[1], Malte Roerden[2,3,6], Jens Bauer[1,2,3], Jonas Rieth[1,3], Marcel Wacker[1,2,3], Sebastian Hörber[7], Andreas Peter[7], Christoph Meisner[8,9], Imma Fischer[8], Markus W. Löffler[2,3,5,10,11], Julia Karbach[12], Elke Jäger[12], Reinhild Klein[6], Hans-Georg Rammensee[2,3,10], Helmut R. Salih[1,2] & Juliane S. Walz[1,2,3,4,13 ✉]

T cell immunity is central for the control of viral infections. CoVac-1 is a peptide-based vaccine candidate, composed of SARS-CoV-2 T cell epitopes derived from various viral proteins[1,2], combined with the Toll-like receptor 1/2 agonist XS15 emulsified in Montanide ISA51 VG, aiming to induce profound SARS-CoV-2 T cell immunity to combat COVID-19. Here we conducted a phase I open-label trial, recruiting 36 participants aged 18–80 years, who received a single subcutaneous CoVac-1 vaccination. The primary end point was safety analysed until day 56. Immunogenicity in terms of CoVac-1-induced T cell response was analysed as the main secondary end point until day 28 and in the follow-up until month 3. No serious adverse events and no grade 4 adverse events were observed. Expected local granuloma formation was observed in all study participants, whereas systemic reactogenicity was absent or mild. SARS-CoV-2-specific T cell responses targeting multiple vaccine peptides were induced in all study participants, mediated by multifunctional T helper 1 CD4+ and CD8+ T cells. CoVac-1-induced IFNγ T cell responses persisted in the follow-up analyses and surpassed those detected after SARS-CoV-2 infection as well as after vaccination with approved vaccines. Furthermore, vaccine-induced T cell responses were unaffected by current SARS-CoV-2 variants of concern. Together, CoVac-1 showed a favourable safety profile and induced broad, potent and variant of concern-independent T cell responses, supporting the presently ongoing evaluation in a phase II trial for patients with B cell or antibody deficiency.

The coronavirus disease 2019 (COVID-19) pandemic caused by severe acute respiratory syndrome coronavirus 2 (SARS-CoV-2) is linked to the death of millions of people[3]. As predominantly individuals with medical comorbidities are severely affected[4], vaccines inducing long-lasting immunity, particularly in high-risk populations, are needed[5–7].

CoVac-1 is a multi-peptide-based vaccine candidate designed to induce, upon a single vaccination, a broad and long-lasting SARS-CoV-2 T cell immunity resembling that acquired by natural infection, which is not affected by evolving viral variants of concern (VOCs). Thus, CoVac-1 is composed of multiple SARS-CoV-2 HLA-DR T cell epitopes derived from various viral proteins (spike, nucleocapsid, membrane, envelope and open reading frame 8 (ORF8)) that have been proven to be (1) frequently and HLA-independently recognized by T cells in

convalescent individuals after COVID-19, (2) of pathophysiological relevance for T cell immunity to combat COVID-19, and (3) to mediate long-term immunity after infection[1,2]. CoVac-1 vaccine peptides are adjuvanted with the novel Toll-like receptor (TLR) 1/2 agonist XS15 emulsified in Montanide ISA51 VG, which endorse activation and maturation of antigen-presenting cells and prevent vaccine peptides from immediate degradation, enabling the induction of a potent T cell response[8–10].

T cells have an important role for COVID-19 outcome and maintenance of SARS-CoV-2 immunity, even in the absence of humoral immune responses[1,11–19]. Thus, the induction of SARS-CoV-2 T cell immunity is a central goal for vaccine development and of particular importance for patients with congenital or acquired B cell deficiencies. The latter comprise patients with cancer or treatment-related immunoglobulin

[1]Clinical Collaboration Unit Translational Immunology, German Cancer Consortium (DKTK), Department of Internal Medicine, University Hospital Tübingen, Tübingen, Germany. [2]Cluster of Excellence iFIT (EXC2180) "Image-Guided and Functionally Instructed Tumor Therapies", University of Tübingen, Tübingen, Germany. [3]Institute for Cell Biology, Department of Immunology, University of Tübingen, Tübingen, Germany. [4]Dr. Margarete Fischer-Bosch Institute of Clinical Pharmacology, Stuttgart, Germany. [5]Department of Clinical Pharmacology, University Hospital Tübingen, Tübingen, Germany. [6]Department of Hematology, Oncology, Clinical Immunology and Rheumatology, University Hospital Tübingen, Tübingen, Germany. [7]Institute for Clinical Chemistry and Pathobiochemistry, Department for Diagnostic Laboratory Medicine, University Hospital Tübingen, Tübingen, Germany. [8]Institute for Clinical Epidemiology and Applied Biometry, University Hospital Tübingen, Tübingen, Germany. [9]Robert Bosch Hospital, Robert Bosch Society for Medical Research, Stuttgart, Germany. [10]German Cancer Consortium (DKTK) and German Cancer Research Center (DKFZ), partner site Tübingen, Tübingen, Germany. [11]Department of General, Visceral and Transplant Surgery, University Hospital Tübingen, Tübingen, Germany. [12]Department of Oncology and Hematology, Krankenhaus Nordwest, Frankfurt, Germany. [13]Robert Bosch Center for Tumor Diseases (RBCT), Stuttgart, Germany. [14]These authors contributed equally: Jonas S. Heitmann, Tatjana Bilich, Claudia Tandler. ✉e-mail: juliane.walz@med.uni-tuebingen.de

deficiency, who develop only limited humoral immunity after infection or vaccination and persist with a high risk for a severe course of COVID-19[20–22].

Here we report the results of the open-label first-in-human phase I trial recruiting adults aged 18–80 years, to evaluate the safety, reactogenicity and immunogenicity of CoVac-1.

## Participants

From 28 November 2020 to 15 January 2021, 12 healthy adults were enrolled in part I (age group 18–55 years), including sentinel dosing in the first participant. From 24 March 2021 to 1 April 2021, 24 adults were enrolled in part II (age group 56–80 years). Of part I and part II participants, 33% and 50%, respectively, were female participants. The median participant age was 38 (range 23–50) and 62 (range 56–70) years for part I and part II, respectively. All participants (pCoVs) received one dose of CoVac-1 on day 1 and were available for immunogenicity and safety analyses until day 28 (follow-up until month 3) and day 56, respectively (Extended Data Fig. 1). No major protocol violations occurred. Analyses of follow-up safety and long-term immunogenicity data (until month 6) are ongoing. Demographic and clinical characteristics of the participants are provided in Table 1.

## Safety and reactogenicity

Data regarding solicited and unsolicited adverse events were available for all participants from diary cards (for 28 days after vaccination) and safety visits (until day 56). No participant discontinued the trial because of an adverse events. No serious adverse events and no grade 4 adverse events were reported. Reactogenicity in terms of solicited adverse events occurred in all participants (Fig. 1). Events were mild to moderate (grade 1–2) in 81% of participants. All participants showed expected formation of an induration (also called granuloma) at the injection site, which persisted beyond day 56. Severe adverse events (grade 3) comprised local erythema in 19%, accompanied by severe swelling in 6% of all participants. Grade 3 adverse events resolved within 2 days (median, range 1–7). Localized inguinal lymphadenopathy was reported by 22% of participants. Local skin ulceration at the vaccination site was reported by 25% of participants, with two participants in part II showing a grade 2 ulceration. Ulcerations in terms of small skin defects occurred between day 28 and day 56 and healed within 20 days (median, range 15–23) until day 56, none requiring any surgical intervention or drug treatment. No difference in local solicited adverse events was observed between part I and part II participants (Extended Data Table 1). No fever or other inflammatory systemic solicited adverse events were reported. Other systemic solicited adverse events occurred in 39% of all participants with no differences observed between part I and part II participants (Extended Data Table 1). All reported systemic solicited adverse events were mild, with transient fatigue being reported by 31% of participants.

No clinically relevant changes in laboratory values were reported. In 31% of participants, acute phase reaction with elevated levels of C-reactive protein was observed.

Fifty-eight unsolicited adverse events occurred that were predominantly mild (81%; Extended Data Table 2). Viral re-activations (varicella zoster and herpes simplex virus) were reported by two participants (grade 2 or lower) in part II of the trial.

Until day 56, no SARS-CoV-2 infection or immune-mediated medical condition was observed in any participant.

## Immunogenicity

Immunogenicity of CoVac-1 was determined in terms of CD4[+] and CD8[+] T cell responses to the six SARS-CoV-2 HLA-DR vaccine T cell epitopes as well as to embedded HLA class I-binding peptides (Supplementary Table 1) using IFNγ enzyme-linked immunospot (ELISPOT)

**Table 1 | Characteristics of participants**

| Characteristics | All | Part I | Part II |
|---|---|---|---|
| Participants; $n$ | 36 | 12 | 24 |
| Age; years | | | |
| Median | 59.5 | 38.0 | 62.0 |
| Range | 23–70 | 23–50 | 56–70 |
| Mean (s.d.) | 54.8 (12.9) | 38.7 (8.2) | 62.8 (4.1) |
| Sex; $n$ (%) | | | |
| Female | 16 (44) | 4 (33) | 12 (50) |
| Male | 20 (56) | 8 (67) | 12 (50) |
| Ethnicity; $n$ (%) | | | |
| White | 36 (100) | 12 (100) | 24 (100) |
| Other | 0 (0) | 0 (0) | 0 (0) |
| Body mass index[a] | | | |
| Median | 24.4 | 24.9 | 24.4 |
| Range | 18.5–30.1 | 20.1–30.1 | 18.5–29.3 |
| Relevant pre-existing disease[b]; $n$ (%) | | | |
| Hypertension | 6 (16.7) | 0 (0) | 6 (25) |
| Previous malignant disease | 1 (2.8) | 0 (0) | 1 (4.2) |
| Mild psoriasis | 1 (2.8) | 0 (0) | 1 (4.2) |

[a]Weight in kg m$^{-2}$; assessment was done at the time of screening.
[b]Relevant pre-existing disease includes conditions with increased risk of severe COVID-19 and with higher risk for CoVac-1 side effects.

assays. T cell responses were assessed in all participants at baseline (day 1), on days 7, 14 and 28, as well as in the follow-up period on day 56 and month 3 after vaccination. None of the participants showed pre-existing SARS-CoV-2 T cell responses ex vivo at baseline. Vaccine-induced IFNγ T cell responses were observed in 100% of participants in part I and part II on day 28, showing a 200-fold or more and 100-fold or more increase (median calculated spot counts 2 (day 1) to 450 (day 28), and 2 (day 1) to 325 (day 28)) from baseline, respectively (Fig. 2a). Vaccine-induced T cell responses targeted multiple CoVac-1 peptides with a median 5 out of 6 peptides recognized by T cells of participants on day 28 (Fig. 2b, Extended Data Fig. 2). The CoVac-1 peptide P6_ORF8 derived from the ORF8 of SARS-CoV-2 showed most frequently induced T cell responses after vaccination (97%), followed by P5_mem and P4_env (both 94%), P3_spi (89%), P1_nuc (61%) and P2_nuc (58%; Extended Data Fig. 2). CoVac-1-induced T cell responses persisted in the follow-up analyses until month 3 in all participants. Intensity of IFNγ T cell response decreased ex vivo in part I participants over time, but equivalent expandability of CoVac-1-induced T cells was observed in both part I and part II participants, at month 3 compared with day 28 post-vaccination (Extended Data Fig. 3a). The intensity of CoVac-1-induced IFNγ T cell responses in participants of part I and part II at day 28 and day 56 (pCoVs ($n = 24$), median 488 and 319 calculated spot counts, respectively) was up to 39 times higher than T cell responses against CoVac-1 vaccine peptides (median 13), as well as to previously described SARS-CoV-2-specific (median 29) and cross-reactive (median 35) T cell epitopes[1,2] in age-matched human convalescent individuals after COVID-19 collected 16–52 days after positive SARS-CoV-2 real-time PCR (Fig. 2c, Supplementary Table 2). Titration with decreasing peptide concentrations (2.5 μg ml$^{-1}$ to 0.1 ng ml$^{-1}$) revealed detection of CoVac-1 peptides by vaccine-induced T cells down to 1 ng ml$^{-1}$ (10 ng ml$^{-1}$ for 5 out of 5 pCoVs, 1 ng ml$^{-1}$ for 3 out of 5 pCoVs). This was lower than the detection limits of SARS-CoV-2-specific T cells in human convalescent individuals for CoVac-1 vaccine peptides (10 ng ml$^{-1}$ for 4 out of 5 human convalescent individuals, and 1 ng ml$^{-1}$ for 0 out of 5 human convalescent individuals), SARS-CoV-2-specific

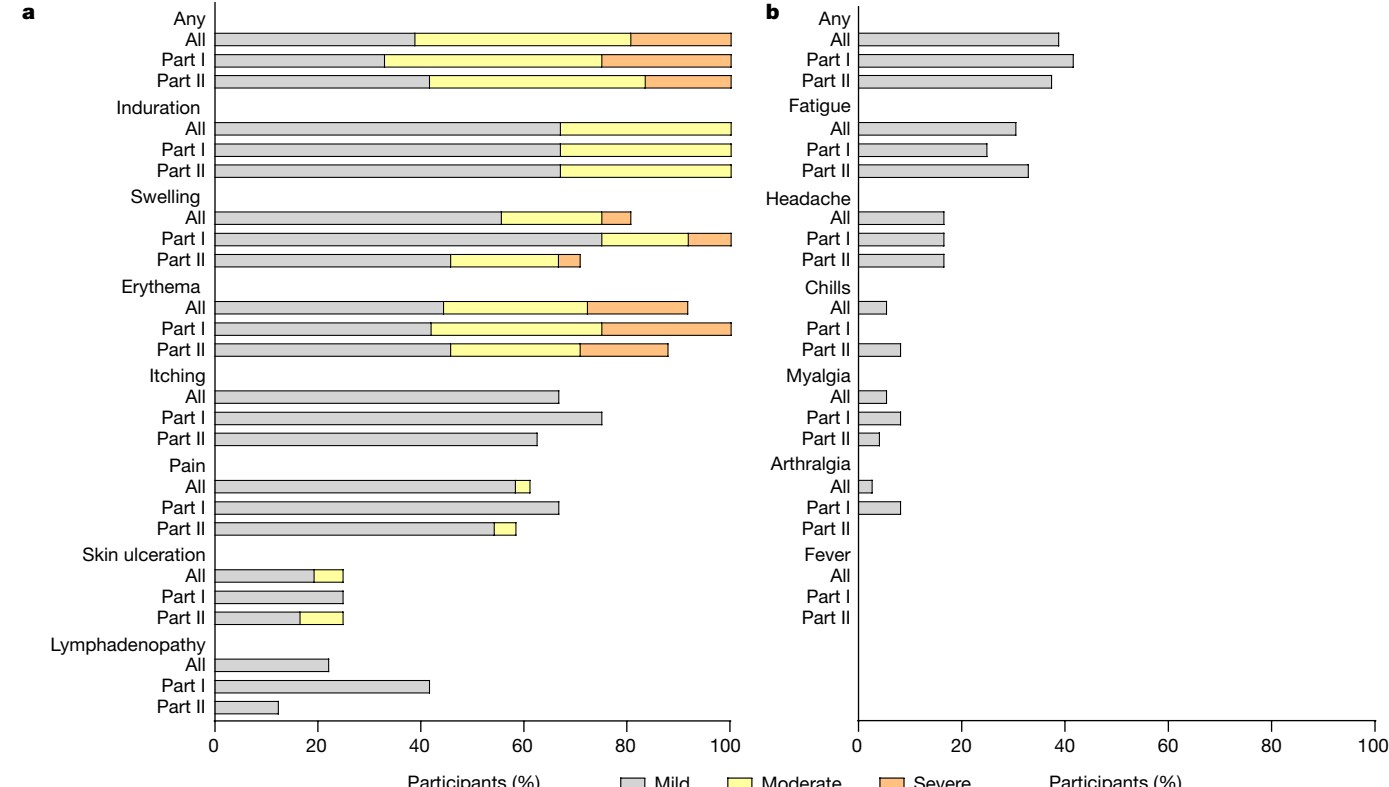

**Fig. 1 | Local and systemic solicited adverse events. a, b**, Related local (**a**) and systemic (**b**) solicited adverse events within 56 days after vaccination. Severity was graded as mild (grade 1), moderate (grade 2) or severe (grade 3) based on the definition provided in Methods. Healthy adults 18–55 years of age were included in part I (n = 12), and participants 56–80 years of age were included in part II (n = 24). A detailed description of the data is presented in Extended Data Table 1.

(10 ng ml$^{-1}$ for 5 out of 5 human convalescent individuals, and 1 ng ml$^{-1}$ for 0 out of 5 human convalescent individuals) and cross-reactive T cell epitopes (10 ng ml$^{-1}$ for 2 out of 5 human convalescent individuals, and 1 ng ml$^{-1}$ for 0 out of 5 human convalescent individuals; Extended Data Fig. 3b). The intensity of CoVac-1-induced IFNγ T cell responses (pCoVs, median of 488 calculated spot counts) exceeded spike-specific T cell responses induced by mRNA-based (median 141), adenoviral vector-based (median 24) and heterologous (median 98) vaccination assessed 18–42 days after the second vaccination (Extended Data Fig. 3c, Supplementary Table 3).

In vitro expansion of CoVac-1-specific T cells revealed pre-existing low-frequency T cell responses to single-vaccine peptides at baseline in 61% of participants that could be boosted at least twofold by CoVac-1, as observed on day 28 in all but one participant (Extended Data Fig. 4).

CoVac-1-induced CD4$^+$ T cells displayed a multifunctional T helper 1 (T$_H$1) phenotype with positivity for IFNγ, tumour necrosis factor (TNF), interleukin-2 (IL-2) and CD107a (Fig. 2d). The magnitude of CoVac-1-induced CD4$^+$ T cell responses did not differ between part I and part II participants and was up to 40 times higher than SARS-CoV-2-specific CD4$^+$ T cell responses of human convalescent individuals (0.42% versus 0.01% (median positive samples) CoVac-1-specific IFNγ$^+$CD4$^+$ T cells in part II participants versus human convalescent individuals, respectively; Fig. 2d, Extended Data Fig. 5a). The frequency of functional CD4$^+$ T cells was increased up to 40-fold after in vitro expansion (17.9% versus 0.44% (median positive samples) of CoVac-1-specific TNF$^+$CD4$^+$ T cells in part I participants), reaching up to 15 times higher levels than expanded CoVac-1-specific T cells of human convalescent individuals (18.6% versus 1.23% (median positive samples) CoVac-1-specific TNF$^+$CD4$^+$ T cells in part II participants versus human convalescent individuals, respectively), indicating potent expandability of CoVac-1-induced T cells upon SARS-CoV-2 exposure (Extended Data Fig. 5b, c).

Vaccine-induced CD8$^+$ T cell responses, identified after in vitro expansion by tetramer staining and IFNγ ELISPOT assay with HLA-matched, CoVac-1-embedded, HLA class I peptides (Supplementary Table 1) were detected in 78% and 80% of participants in part I and 100% and 95% of participants in part II with matching HLA allotypes, respectively (Extended Data Fig. 6a, b). CoVac-1-induced CD8$^+$ T cells showed a polyfunctional phenotype reflected by IFNγ, TNF, IL-2 and CD107a production or expression (Extended Data Fig. 6c).

No relevant differences were observed for immunogenicity parameters between part I and part II participants except for the frequency of IL-2$^+$ CoVac-1-specific CD4$^+$ T cells following 12-day in vitro expansion at day 28, which was increased in part II participants, and for the expandability of CoVac-1-induced T cells at the follow-up time points (day 56 and month 3), which was decreased in part II compared with part I participants (Extended Data Table 3).

In addition to T cell responses, the induction of low-concentration SARS-CoV-2 anti-spike IgG antibodies could be observed in two participants on day 28 (Extended Data Fig. 3d).

## Impact of SARS-CoV-2 variants on CoVac-1

The impact of SARS-CoV-2 VOCs declared by the World Health Organization as of 1 October 2021 (B.1.1.7 (also known as Alpha), B.1.351 (also known as Beta), P.1 (also known as Gamma) and B.1.617.2 (also known as Delta)) on CoVac-1 was analysed comparing CoVac-1 peptides with the corresponding mutated regions of the respective source proteins described for each VOC (Supplementary Table 4). The sequences of 50% of vaccine peptides were not affected by any variant-defining or associated mutation[23–26] (Supplementary Table 4). None of the mutations of P.1 and B.1.617.2 affect CoVac-1 vaccine peptides. Variant B.1.1.7 comprises two mutations affecting P2_nuc and P6_ORF8 with a single

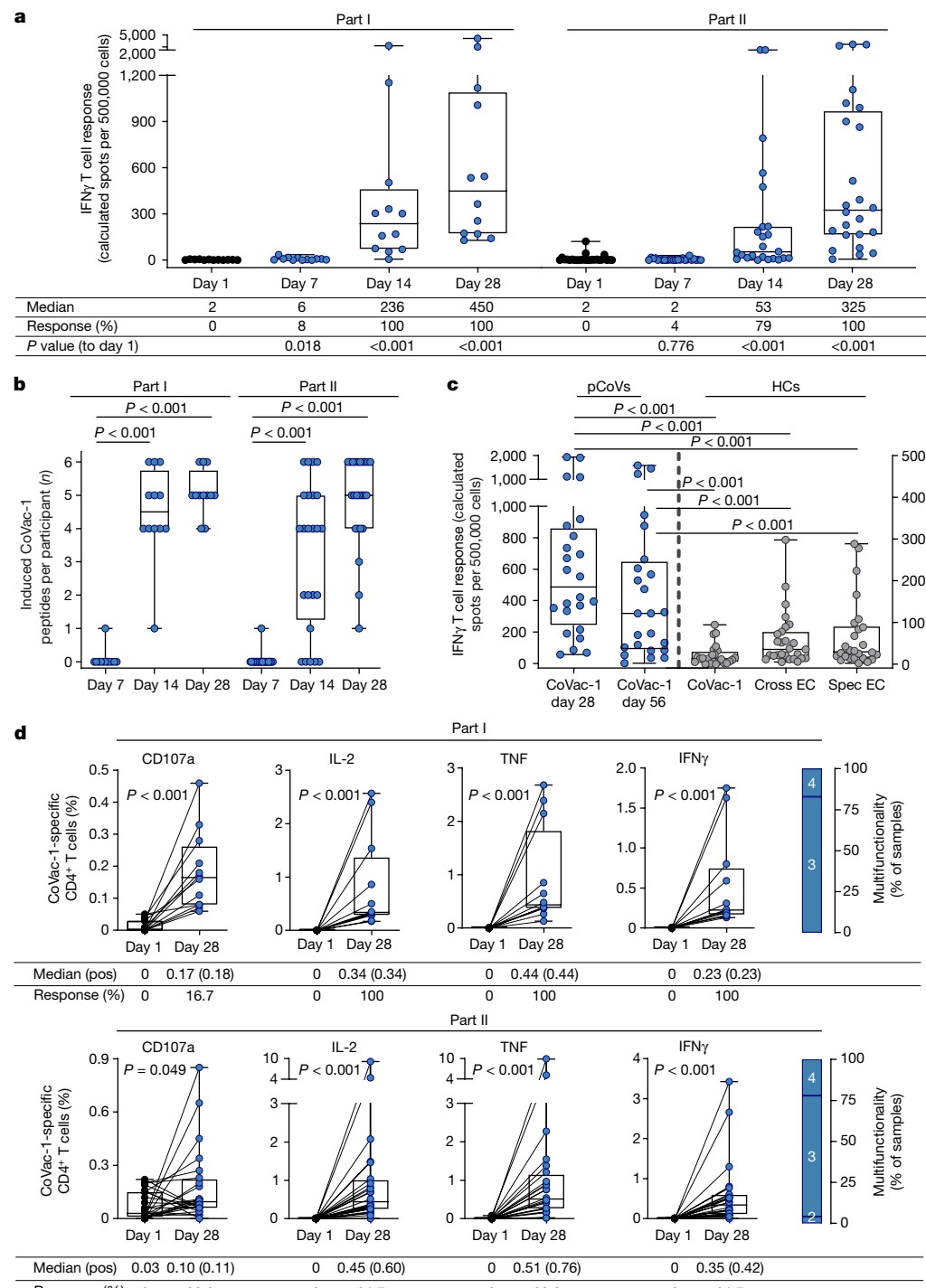

**Fig. 2 | CoVac-1-induced T cell responses. a–c**, CoVac-1-induced T cell responses assessed ex vivo by IFNγ ELISPOT assays using peripheral blood mononuclear cells from study participants of part I (*n* = 12) and part II (*n* = 24) collected before vaccination (day 1) and at different time points after vaccination (days 7, 14, 28 and 56) or from human convalescent individuals (HCs). The intensity of T cell responses is depicted as cumulative calculated spot counts (mean spot count of technical replicates normalized to 500,000 cells minus the respective negative control) (**a**). The number of CoVac-1 T cell epitopes (*n* = 6) per participant that elicited a vaccine-induced T cell response (**b**). Intensities of CoVac-1-induced IFNγ T cell responses assessed ex vivo in part I and part II study participants (pCoVs; *n* = 24, day 28 and day 56, left *y* axis) compared with T cell responses detected in HCs (right *y* axis) against CoVac-1 vaccine peptides and previously published[1,2] SARS-CoV-2-specific (spec) and cross-reactive (cross) T cell epitope compositions (ECs; CoVac-1 *n* = 24, cross EC *n* = 27, spec EC *n* = 26) (**c**). **d**, Frequencies of functional CoVac-1-induced CD4+ T cells in study participants before vaccination (day 1) and at day 28 following vaccination using ex vivo intracellular cytokines (IFNγ, TNF and IL-2) and surface marker staining (CD107a). The right graph displays the proportion of samples revealing difunctional (2), trifunctional (3) or tetrafunctional (4) T cells. Pos, positive. In **a–d**, the box plots or combined box-line plots show the median with 25th or 75th percentiles, and minimum and maximum whiskers. In **a**, **b**, **d**, two-sided Wilcoxon signed-rank test was used; in **c**, two-sided Mann–Whitney *U*-test was used. Healthy adults 18–55 years of age were included in part I, and participants 56–80 years of age were included in part II. pos, positive.

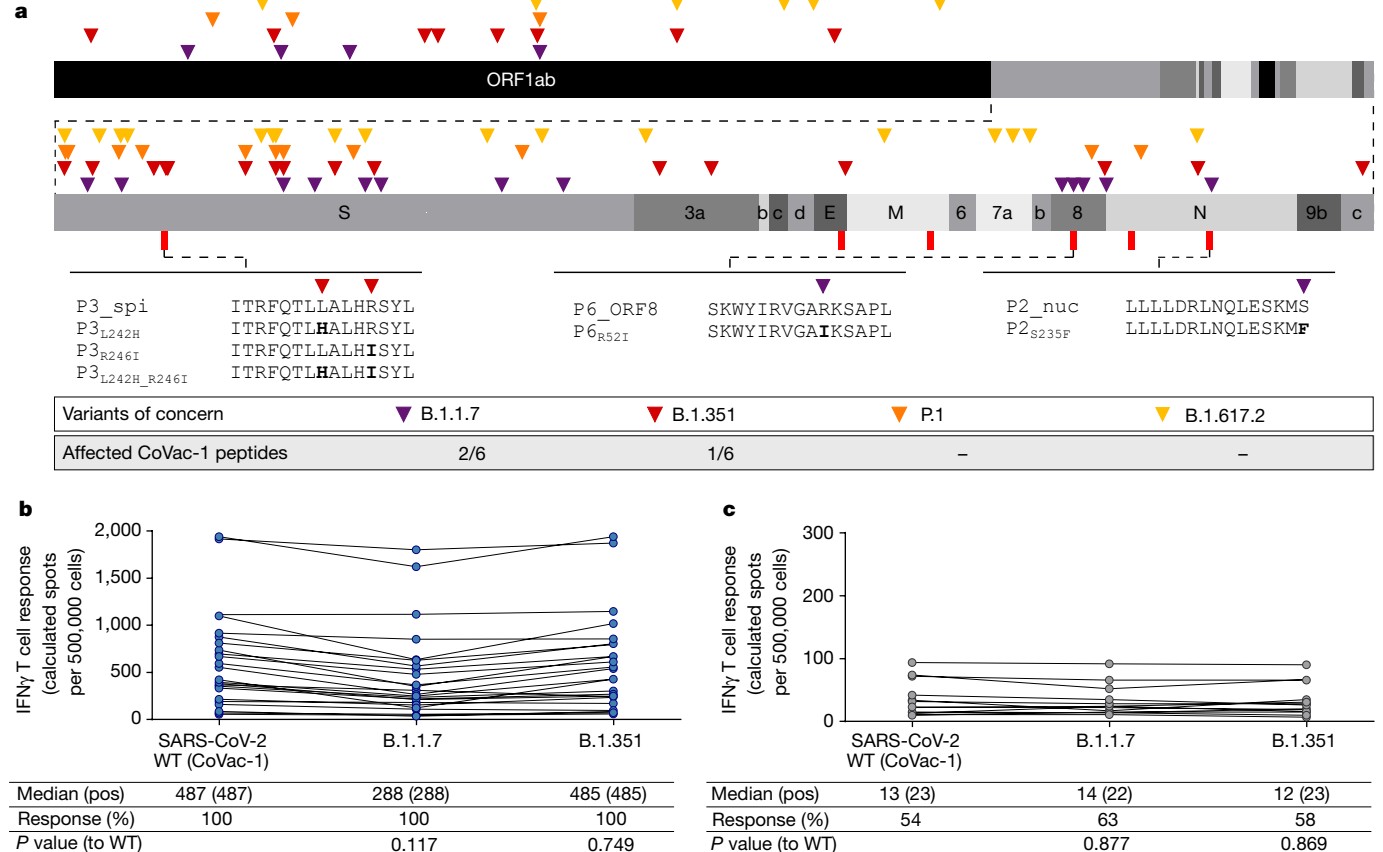

| P3_spi | ITRFQTLLALHRSYL |
| P3_L242H | ITRFQTL**H**ALHRSYL |
| P3_R246I | ITRFQTLLALH**I**SYL |
| P3_L242H_R246I | ITRFQTL**H**ALH**I**SYL |

| P6_ORF8 | SKWYIRVGARKSAPL |
| P6_R52I | SKWYIRVGA**I**KSAPL |

| P2_nuc | LLLLDRLNQLESKMS |
| P2_S235F | LLLLDRLNQLESKM**F** |

| Variants of concern | ▼ B.1.1.7 | ▼ B.1.351 | ▼ P.1 | ▼ B.1.617.2 |
|---|---|---|---|---|
| Affected CoVac-1 peptides | 2/6 | 1/6 | – | – |

**b**

| | SARS-CoV-2 WT (CoVac-1) | B.1.1.7 | B.1.351 |
|---|---|---|---|
| Median (pos) | 487 (487) | 288 (288) | 485 (485) |
| Response (%) | 100 | 100 | 100 |
| *P* value (to WT) | | 0.117 | 0.749 |

**c**

| | SARS-CoV-2 WT (CoVac-1) | B.1.1.7 | B.1.351 |
|---|---|---|---|
| Median (pos) | 13 (23) | 14 (22) | 12 (23) |
| Response (%) | 54 | 63 | 58 |
| *P* value (to WT) | | 0.877 | 0.869 |

**Fig. 3 | Role of SARS-CoV-2 variants of concern on CoVac-1 peptides and immunogenicity. a**, Colour-coded mutations described for variants of concern are shown together with corresponding affected CoVac-1 peptides. **b**, **c**, Intensities of T cell responses (calculated spot counts) to CoVac-1 peptides as well as to the corresponding peptide pools comprising the CoVac-1-affecting amino acid change, respectively. Two mutations of B.1.351 affect P3_spi with either one or two amino acid changes (Fig. 3a). mutations of B.1.1.7 and B.1.351 were assessed ex vivo by IFNγ ELISPOT assays using peripheral blood mononuclear cells from study participants of part I (n = 12) and part II (n = 24) collected on day 28 after vaccination (pCoVs) (**b**) or from HCs (**c**). Two-sided Mann–Whitney U-test was used.

amino acid change, respectively. Two mutations of B.1.351 affect P3_spi with either one or two amino acid changes (Fig. 3a).

T cell responses to peptide pools comprising the B.1.1.7 and B.1.351 mutated peptides P2_nuc, P3_spi and P6_ORF8 were detectable in 100% of part I and part II participants with CoVac-1-induced T cell responses to P2_nuc, P3_spi and P6_ORF8 wild-type (WT) peptides (Fig. 3b). Although the intensity of T cell responses to single-peptide variants (P3_spi and P6_ORF8) was reduced compared with WT peptides, the intensity of CoVac-1-induced T cell responses targeting the variant peptide pools was unaffected and at least 10-fold higher than T cell responses to WT and variant peptide pools observed in human convalescent individuals (median calculated spot counts 288 pCoVs B.1.1.7, 485 pCoVs B1.351, 13 WT human convalescent individuals, 14 B1.1.7 human convalescent individuals and 12 B.1.351 human convalescent individuals; Fig. 3c, Extended Data Fig. 7).

## Discussion

Our phase I trial shows that the CoVac-1 vaccine candidate has a favourable safety profile and induces potent T cell responses after a single vaccination. Local granuloma formation was observed in all study participants representing an expected and intended local reaction after Montanide-based vaccination[9,27], which enables continuous local stimulation of SARS-CoV-2-specific T cells required for induction of long-lasting T cell responses without systemic inflammation. Follow-up data until month 3 after vaccination showed persistence of T cell responses, which is in line with previous experience with XS15-adjuvanted peptide vaccinations[8] and data from SARS-CoV-1 convalescent individuals, where T cell immunity persisted for up to 17 years[16]. CoVac-1-induced $T_H1$ CD4[+] T cell responses were complemented by multifunctional CD8[+] T cells, counteracting the theoretical risk of vaccine-associated enhanced respiratory disease, which has been associated with a $T_H2$-driven immune response[28]. The phenotype of CoVac-1-induced T cells resembles that acquired upon natural infection[1,2,11,16], but with higher magnitude than the SARS-CoV-2 T cell responses in human convalescent individuals as well as than spike-specific T cell responses induced by mRNA-based, vector-based and heterologous vaccination[5–7,29], substantiating the profound T cell immunity induced by CoVac-1. This is further supported by the high diversity of CoVac-1-induced T cells that target multiple vaccine peptides from different viral proteins, which is central for effective anti-viral defence[1,30–32]. These broad T cell responses induced by CoVac-1 remain unaffected by current SARS-CoV-2 VOCs, which were associated with loss of neutralizing antibody capacity in convalescent individuals after COVID-19 and after vaccination[33–35].

In single participants, despite negative results in sequential SARS-CoV-2 PCRs, induction of SARS-CoV-2 anti-spike IgG antibodies was documented after vaccination. This might be due to CoVac-1-induced profound CD4[+] T cell responses, which not only stimulate B cells upon virus encounter but also may boost pre-existing cross-reactive SARS-CoV-2 antibodies, which were reported in 3–15% of unexposed individuals[36].

T cell-mediated immunity and in particular CD4[+] T cells are indispensable for the generation of protective antibody responses,

reinforcement of CD8[+] T cell responses[37–39] as well as direct killing of virus-infected cells[40,41]. The relevance of anti-viral T cell responses during acute infection and for long-term immunity has also been proven specifically for SARS-CoV-2[1,2,13–19]. Moreover, cases of asymptomatic SARS-CoV-2 exposure, as well as reports from patients with congenital B cell deficiency document cellular immune responses without seroconversion, providing evidence for T cell immunity in disease control even in the absence of neutralizing antibodies[14,42]. Accordingly, CoVac-1 may well serve as a (complementary) vaccine to induce T cell immunity, particularly in elderly and immunocompromised individuals with impaired ability to mount sufficient immune responses after SARS-CoV-2 vaccination with currently approved vaccines[20,21].

Limitations of our trial include the small sample size, low ethnic diversity, as well as the non-equivalent time points of sample collection in the comparison of vaccine-induced and infection-induced SARS-CoV-2 T cells.

In conclusion, the safety and immunogenicity results of this trial indicate that CoVac-1 is a promising multi-peptide vaccine candidate for induction of profound SARS-CoV-2 T cell immunity, which builds the basis for a presently ongoing phase II study evaluating CoVac-1 in patients with congenital or acquired B cell defects, including patients with cancer after B cell-depleting therapy and disease-related immunoglobulin deficiency (NCT04954469).

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

# Methods

## Trial design and oversight

The phase I trial (ClinicalTrials.gov identifier: NCT04546841) was designed by and conducted at the Clinical Collaboration Unit (CCU) Translational Immunology, University Hospital Tübingen, Germany. Men as well as nonpregnant women aged 18–55 years, without any relevant pre-existing conditions and adults aged 56–80 years with stable medical conditions were included in part I and part II of the study, respectively. A detailed description of the inclusion and exclusion criteria can be found in the Supplementary Information. Health status was based on medical history and clinical laboratory values, vital signs and physical examination at screening. Participants with a proven history of SARS-CoV-2 infection (real-time PCR or antibody test) were excluded. Before enrolment, all participants provided their written informed consent. As a safety measure, sentinel dosing of the first participant treated in part I was conducted with a follow-up period of 28 days after vaccination followed by a sponsor safety assessment before proceeding with the vaccination of further study participants. Safety assessment of the sentinel dosing participant is described in detail in the Supplementary Information. The trial was open-label (no blinding) without a control arm (no randomization).

The trial was funded by the Ministry of Science, Research and the Arts Baden-Württemberg, Germany. The trial was approved by the Ethics Committee, University Tübingen (537/2020AMG1) and the Paul Ehrlich Institute and performed in accordance with the International Council for Harmonization Good Clinical Practice guidelines.

Safety assessment to proceed to part II was performed by an independent data safety monitoring board (DSMB).

## Trial vaccine and adjuvant

CoVac-1, developed and produced by the Good Manufacturing Practices (GMP) Peptide Laboratory of the Department of Immunology, University Tübingen, is a peptide-based vaccine comprising six HLA-DR-restricted SARS-CoV-2 peptides (Supplementary Table 1) derived from various SARS-CoV-2 proteins (spike, nucleocapsid, membrane, envelope and ORF8) and the adjuvant lipopeptide synthetic TLR1/2 ligand XS15[8] (manufactured by Bachem AG) emulsified in Montanide ISA51 VG[9] (manufactured by Seppic). CoVac-1 peptides represent dominant SARS-CoV-2 T cell epitopes (peptide-specific T cell responses detected in more than 50% and up to 100% of convalescent individuals after SARS-CoV-2 infection) validated in human convalescent individuals after SARS-CoV-2 infection to mediate long-term immunity[1,2]. CoVac-1 peptides were predicted and validated to bind to multiple HLA-DR molecules (promiscuous binding)[1], which is important to enable HLA-independent induction of T cell responses by CoVac-1[1,2,43].

CoVac-1 HLA-DR T cell epitopes contain embedded HLA class I sequences for induction of both CD4[+] and CD8[+] T cell responses (Supplementary Table 1). CoVac-1 peptides were selected from viral non-surface proteins and their subunits or—in case of the spike protein-derived T cell epitope P3_spi—from buried (or hidden) amino acid sequences, which are not accessible for antibodies in their conformational state. The linear 15-amino acid peptides are characterized by a free N-terminal amino group and a free C-terminal carboxy group. All amino acid residues are in the L-configuration and were not chemically modified at any position. Synthetic peptides were manufactured by established solid-phase peptide synthesis procedures using Fmoc chemistry[44,45].

The novel adjuvant XS15 hydrochloride is a water-soluble synthetic linear, nine amino acid peptide with a palmitoylated N terminus (Pam$_3$Cys-GDPKHPKSF)[8]. Acting as a TLR1/2 ligand, XS15 strongly activates antigen-presenting cells[8] and enables the induction of strong ex vivo CD8[+] and T$_H$1 CD4[+] responses to viral peptides, including SARS-CoV-2 T cell epitopes, in preliminary in vivo analyses in a human volunteer upon a single subcutaneous injection of XS15 mixed to uncoupled viral peptides in a water-in-oil emulsion with Montanide ISA51 VG[8,10]. To our knowledge, this is the first report of the adjuvant XS15 being used in a human clinical trial. Montanide ISA51 VG is a mixture of a highly purified mineral oil (Drakeol 6VR) and a surfactant (mannide monooleate). When mixed with an aqueous phase in a 50:50 ratio, it forms a water-in-oil emulsion. Such a Montanide-based water-in-oil emulsion has been used as vaccine adjuvant in multiple clinical trials[9,27], to build a depot at the vaccination site, thereby preventing vaccine peptides from immediate degradation and thus enhancing the immune response.

CoVac-1 peptides (250 μg per peptide) and XS15 (50 μg) are prepared as a water–oil emulsion 1:1 with Montanide ISA51 VG to yield an injectable volume of 500 μl. Each participant received one subcutaneous injection of the CoVac-1 vaccine in the lower abdomen on day 1.

The dosage of CoVac-1 vaccine peptides was determined based on results from various clinical trials evaluating peptide vaccines[44,46–51] (including dose-finding studies for viral T cell epitopes), which showed significantly stronger immune responses to 250–500 μg versus 100 μg peptide dose, without significantly higher immune responses in the 1,000 μg versus 500 μg dose group[47]. Similar T cell responses were induced with 250 μg and 500 μg peptide doses. Regarding safety, even doses up to 30 mg per peptide did not raise any concerns[48]. On the basis of these data, the dose of 250 μg per peptide was used for CoVac-1 vaccine peptides.

The dosage of the TLR1/2 agonist XS15 was determined based on in vitro analyses of immune cell activation by TLR1/2. In these assays, 10 μg ml$^{-1}$ XS15 was shown to be the most efficient dose for the stimulation of immune cells. Considering the formation of a granuloma after subcutaneous injection of XS15 emulsified in Montanide ISA51 VG, which leads to a size-dependent decrease in XS15 concentration[8], 50 μg XS15 was selected to achieve the desired dosage of 10 μg ml$^{-1}$ at the vaccination site. In a toxicity study in mice, 50 μg XS15 in Montanide ISA51 VG, applied subcutaneously, did not reveal any local or systemic toxicity beyond the long known and expected local toxicity of Montanide[9,27]. For a more detailed description of the dosage rationale for the vaccine peptides and the adjuvant, please refer to the Supplementary Information.

## Safety assessment

Primary safety outcomes reflect the nature, frequency and severity of solicited adverse events until day 56 after vaccination. The documentation was facilitated by use of a volunteer diary (for 28 days after vaccination) and graded by the investigators according to a modified Common Terminology Criteria for Adverse Events (CTCAE) V5.0 grading scale (Supplementary Table 5). In addition, the number and percentage of participants with unsolicited events until day 56 were reported (documented according to CTCAE V5.0). Safety assessment included clinically significant changes in laboratory values (haematology and blood chemistry), serious adverse events, and adverse events of special interest, which included desired induration (granuloma) formation, SARS-CoV-2 infection, COVID-19 manifestations and immune-mediated medical conditions (Supplementary Tables 6, 7).

## Immunogenicity assessment

Secondary outcome was the induction of CoVac-1-specific T cell responses to at least one of the CoVac-1 vaccine peptides evaluated on day 7, day 14 and day 28 by the IFNγ ELISPOT assay ex vivo and after in vitro T cell expansion (baseline day 1, before vaccination). Follow-up analyses of CoVac-1-induced T cell responses were performed on day 56 and month 3 after vaccination. The 12-day in vitro expansion of peptide-specific T cells was performed to enable detection of low-frequent, vaccine-induced and pre-existing SARS-CoV-2-specific T cells, as well as to prove the expandability of CoVac-1-induced T cells, which is of central importance for potent T cell response upon SARS-CoV-2 exposure. In this regard, the characterization of vaccine-induced CD8[+] T cells was performed after

12-day in vitro expansion, due to the low frequency of peptide-specific CD8$^+$ T cells observed ex vivo (Supplementary Table 8). PBMCs were pulsed with CoVac-1 peptides (5 µg ml$^{-1}$ per peptide) and cultured for 12 days adding 20 U ml$^{-1}$ IL-2 (Novartis) on days 3, 5 and 7. For IFNγ ELISPOT (ex vivo or after in vitro expansion), cells were stimulated with 1 µg ml$^{-1}$ of HLA class I or 2.5 µg ml$^{-1}$ of HLA-DR peptides and analysed in technical replicates. T cell responses were considered positive if the mean spot count was threefold or more higher than the mean spot count of the negative control and defined as CoVac-1-induced if the mean spot count post-vaccination was twofold or more higher than the respective spot count on day 1. CoVac-1-induced T cell responses were further characterized using tetramer (5 µg ml$^{-1}$), cell-surface marker and intracellular cytokine staining. For intracellular cytokine staining, cells were stimulated with 10 µg ml$^{-1}$ per peptide. The gating strategy applied for the analyses of flow cytometry-acquired data is provided in Supplementary Fig. 1. Immunogenicity results were compared with human convalescent individuals with PCR-confirmed SARS-CoV-2 infection and healthy volunteers vaccinated with an approved mRNA-based or vector-based vaccine or received heterologous vaccination (Supplementary Tables 2, 3). All assays were conducted in a blinded manner and are described in detail in the Supplementary Information.

## Statistical analysis

The sample size calculation (36 participants) of the trial was based on the assumption that incidence of serious adverse events associated with administration of CoVac-1 does not exceed a predetermined rate of 5%. Safety data are displayed by counting the respective adverse event that has occurred at least once in a patient. The highest grading of this adverse event is indicated. Data are displayed as mean ± s.d., box plots as median with 25% or 75% quantiles and minimum and maximum whiskers. Continuous data were tested for distribution, and individual groups were tested by use of Fisher's exact test, unpaired Mann–Whitney $U$-test or paired Wilcoxon signed-rank test, all performed as two-sided tests. No adjustment for multiple testing was done. Details regarding the statistical analysis plan and sample size calculation are provided in the Supplementary Information and the protocol.

## Reporting summary

Further information on research design is available in the Nature Research Reporting Summary linked to this paper.

## Data availability

Data supporting the findings of this study including the study protocol and the statistical analysis plan are supplied as source data with this paper. Further data, including de-identified participant data, are available after final completion of the trial report and are shared according to data sharing guidelines on reasonable request to the corresponding author (J.S.W.).

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

**Acknowledgements** We thank all of the participants of this trial and the members of the data safety monitoring board (P. Brossart, Bonn; H. Schild, Mainz; C. Wendtner, Munich); the technical and clinical staff of the CCU Translational Immunology, Department of Internal Medicine, University Hospital Tübingen; the team of the 'Wirkstoffpeptid Labor', Department of Immunology, Tübingen; the pharmacy of the University Hospital Tübingen; the data management team at the Institute for Clinical Epidemiology and Applied Biometry; the 'Zentrum für Klinische Studien' (particularly M. Walker) at the University Hospital Tübingen for support and coordination; and EMC Microcollections GmbH for provision of XS15. This work was supported by the Ministry of Science, Research and the Arts Baden-Württemberg, Germany (Sonderfördermassnahme COVID-19, TÜ17), Bundesministerium für Bildung und Forschung (BMBF, FKZ:01KI20130), the Robert Bosch Stiftung, the Deutsche Forschungsgemeinschaft (DFG, German Research Foundation, grant WA 4608/1-2), the Deutsche Forschungsgemeinschaft under Germany's Excellence Strategy (grant EXC2180-390900677), the German Cancer Consortium (DKTK), the Wilhelm Sander Stiftung (grant 2016.177.2), the José Carreras Leukämie-Stiftung (grant DJCLS 05 R/2017), the Fortüne Program of the University of Tübingen (Fortüne number 2451-0-0) and the Ernst-Jung-Preis awarded to H.-G.R.

**Author contributions** J.S.H., H.-G.R., H.R.S. and J.S.W. were involved in the design of the overall study and strategy. C.M., H.-G.R., I.F. and M.W.L. provided feedback on the study design. T.B., C.T., A.N., Y.M., M.Roerden, J.B., J.Rieth and M.W. performed the immunogenicity analyses. J.S.H., M.M., J.Reusch, S.J., L.A., L.M.W., S.H., A.P., J.K., E.J., R.K. and J.S.W. conducted patient data and sample collection, as well as medical evaluation and analysis. J.S.H., M.M., J.Reusch, S.J., H.R.S. and J.S.W. collected data as study investigators. C.M. and I.F. developed the statistical design and oversaw the data analysis. M.D. and M.Richter conducted GMP production of CoVac-1. J.S.H., T.B., C.T., M.W.L., H.R.S. and J.S.W. drafted the manuscript. All authors supported the review of the manuscript.

**Competing interests** The University Hospital Tübingen is in the process of applying for a patent application (EP 20 169 047.6) covering the SARS-CoV-2 T cell epitopes included in CoVac-1 that lists A.N., T.B., H.-G.R. and J.S.W as inventors. EMC Microcollections GmbH is in the process of applying for a patent application (DE102016005550.2) covering the adjuvant XS15 included in CoVac-1 that lists H.-G.R. as an inventor. The other authors declare no competing interests.

**Additional information**
**Correspondence and requests for materials** should be addressed to Juliane S. Walz.

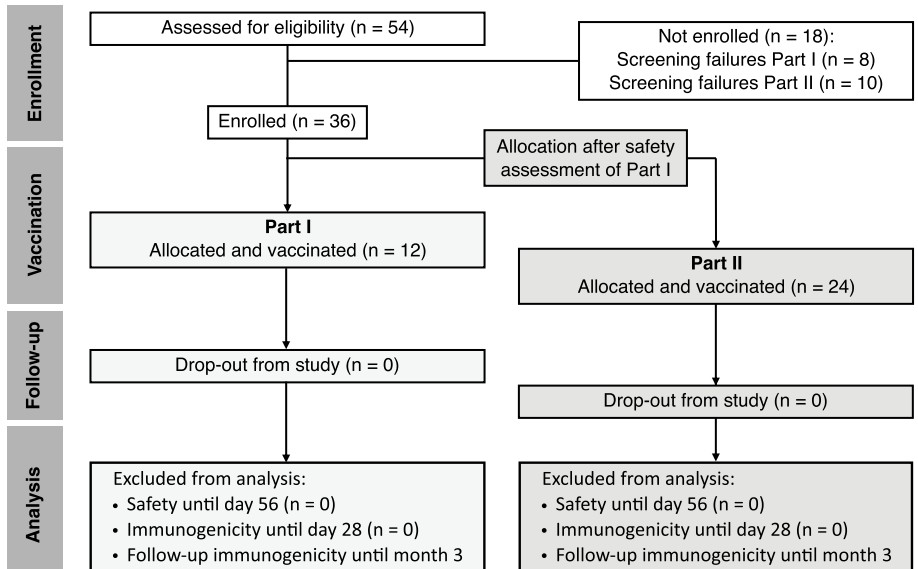

**Extended Data Fig. 1 | Consort flow diagram of the trial.** The 18 participants who were not enrolled did not meet the inclusion criteria at screening. All 36 enrolled participants received one dose of the CoVac-1 vaccine. Safety oversight to proceed to part II was performed by an independent safety monitoring committee and approved by the Paul Ehrlich Institute and the local Ethics Committee after an interim safety and immunogenicity analysis of study participants included in part I on day 28 after vaccine administration. n, number.

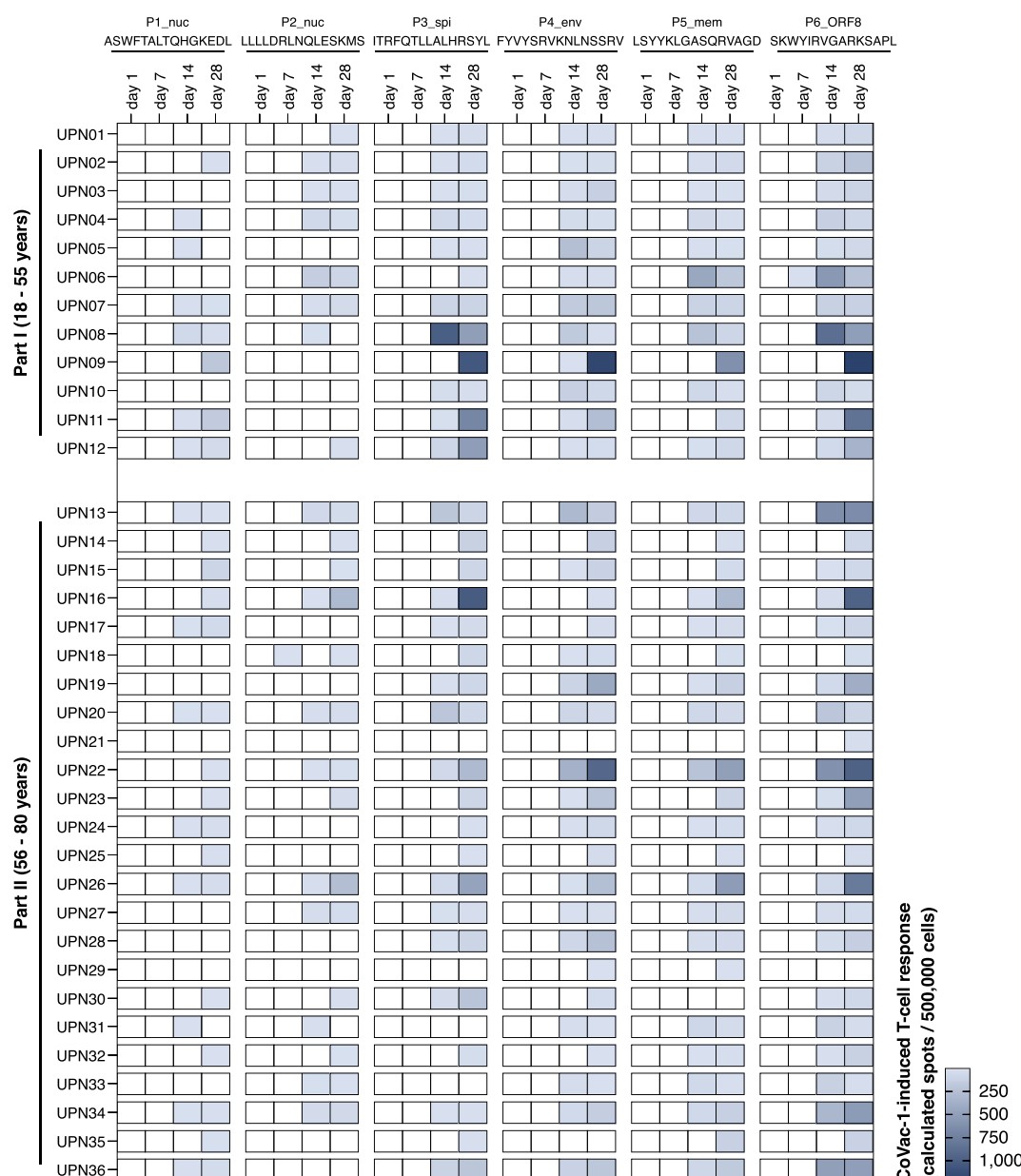

**Extended Data Fig. 2 | Intensities of CoVac-1-induced T cell responses _ex vivo_ assessed in IFNγ ELISPOT assays.** Heatmap of CoVac-1-induced T cell response intensities (calculated spots per 500,000 cells, color gradient blue) to single CoVac-1 peptides (nuc, nucleocapsid; spi, spike; env, envelope; mem, membrane; ORF, open reading frame) in _ex vivo_ IFNγ ELISPOT assays using PBMCs from study participants (uniform participant number, UPN) of part I (n = 12) and part II (n = 24) before vaccination (day 1) and at different time points after vaccination (day 7, day 14, day 28).

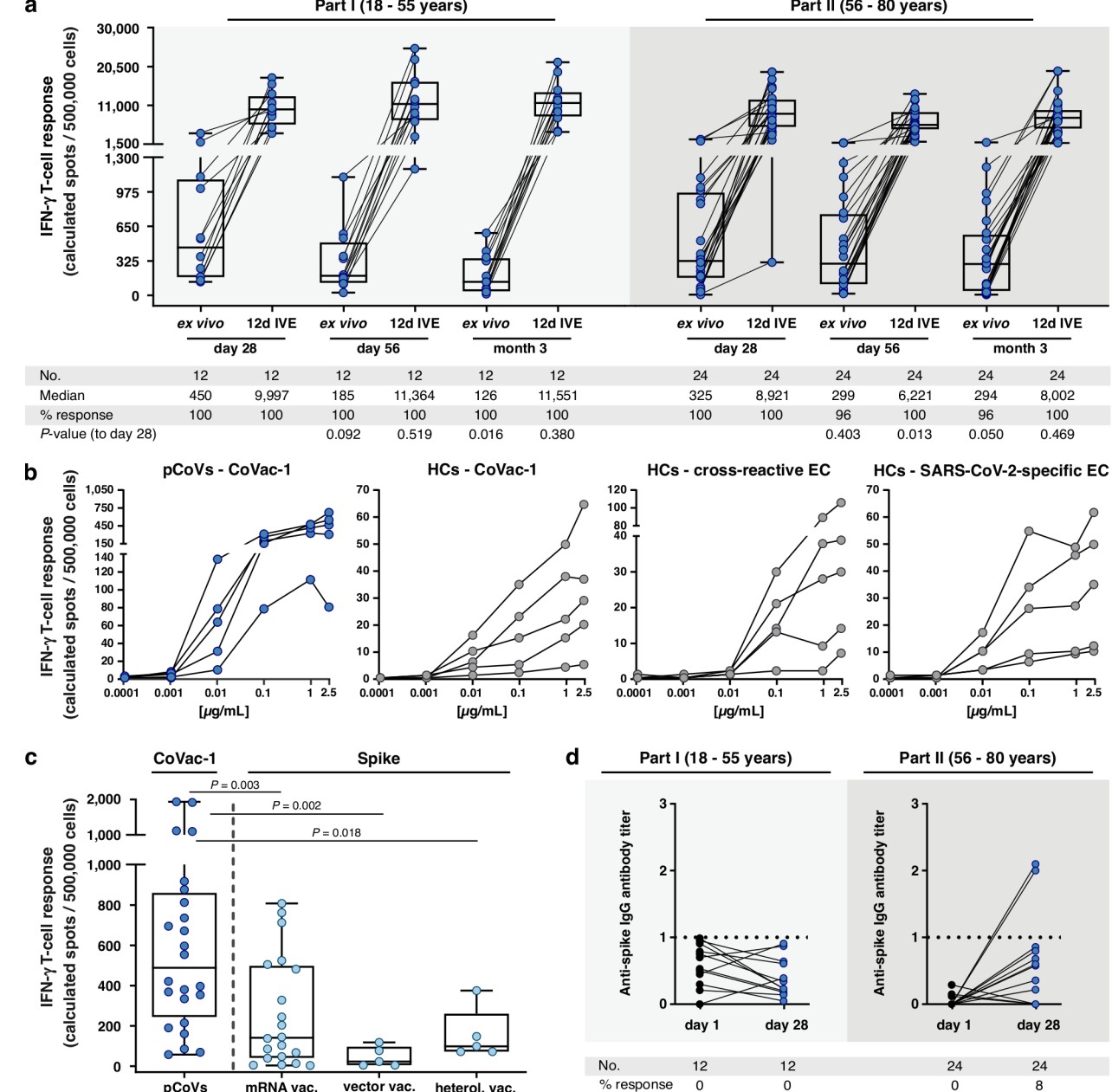

**Extended Data Fig. 3 | Characterization of CoVac-1-induced immune responses. (a)** CoVac-1-induced long-term T cell responses assessed *ex vivo* or after 12-day *in vitro* expansion (IVE) in study participants of part I and II at day 56 and month 3 after vaccination (compared to day 28) using IFNγ ELISPOT assays. Intensity of T cell responses is depicted as calculated spot counts (mean spot count of technical replicates normalized to 500,000 cells minus the respective normalized negative control). **(b)** Peptide titration in *ex vivo* IFNγ ELISPOT assays using PBMCs from study participants (pCoVs, n = 5, day 28) or from human COVID-19 convalescent donors (HCs, n = 5) with decreasing peptide concentrations (2.5 µg mL⁻¹ to 0.1 ng mL⁻¹) of CoVac-1 (panel 1 and 2) or SARS-CoV-2 cross-reactive (panel 3) and SARS-CoV-2 specific (panel 4)

epitope compositions (ECs). **(c)** Intensities of CoVac-1-induced IFNγ T cell responses assessed *ex vivo* in study participants of part I and part II (pCoVs, n = 24, day 28) compared to spike-specific T cell responses in healthy immunized donors after second vaccination with approved mRNA vaccines (n = 20), vector-based vaccines (n = 5), or heterologous vaccination (n = 5). **(d)** Anti-spike IgG antibody titers assessed on day 1 prior to vaccination and on day 28 after vaccination. Values < 0.1 were set to zero and values ≥ 1.0 were considered positive. **(a, c)** Box plots or combined box-line plots show median with 25th or 75th percentiles, and min/max whiskers. **(a)** two-sided Wilcoxon signed-rank test, **(c)** two-sided Mann-Whitney U-test. no, number.

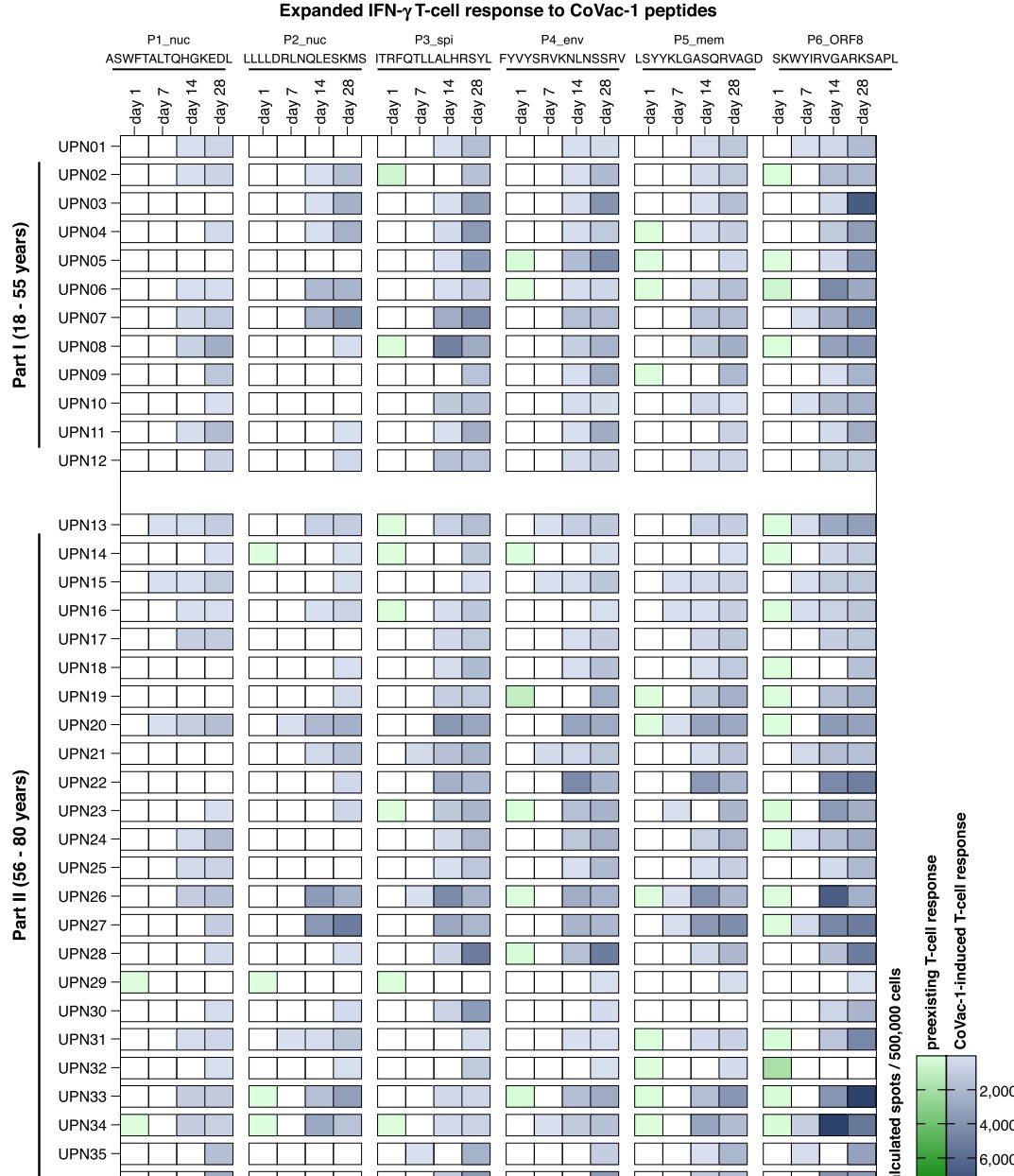

**Extended Data Fig. 4 | Intensities of CoVac-1-induced T cell responses assessed in IFNγ ELISPOT assays after 12-day *in vitro* expansion.** Heatmap of preexisting (color gradient green) or CoVac-1-induced (color gradient blue) T cell response intensities (calculated spots per 500,000 cells) to single CoVac-1 peptides (nuc, nucleocapsid; spi, spike; env, envelope; mem, membrane; ORF, open reading frame) in IFNγ ELISPOT assays after 12-day *in vitro* expansion of PBMCs from study participants (uniform participant number, UPN) of part I (n = 12) and part II (n = 24) before vaccination (day 1) and at different time points after vaccination (day 7, day 14, day 28).

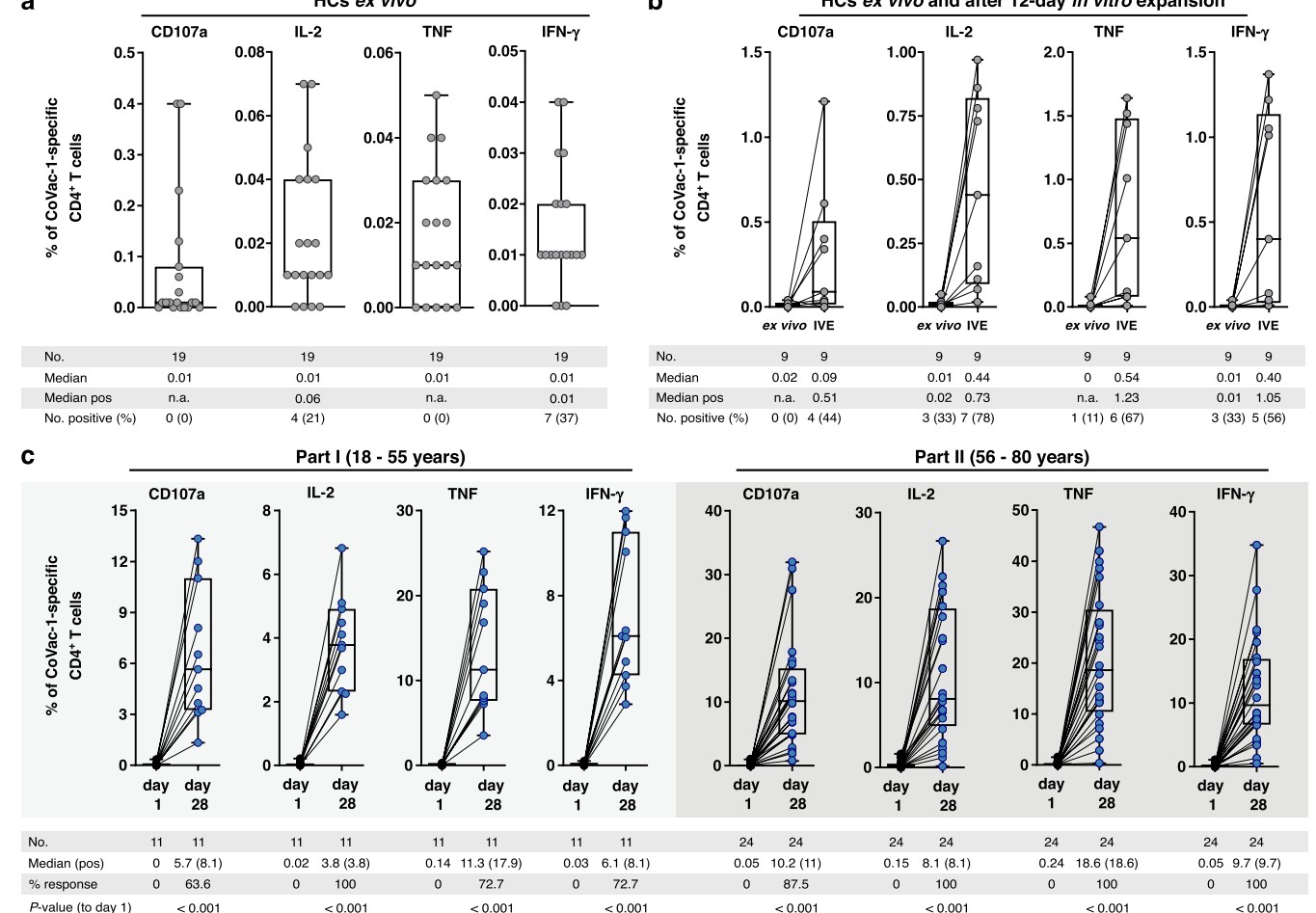

**Extended Data Fig. 5 | CoVac-1-induced CD4⁺ T cell responses in human COVID-19 convalescents and study participants.** (a–c) Frequencies of CoVac-1-specific CD4⁺ T cells in (**a**) human convalescent samples (HCs) after SARS-CoV-2 infection analyzed *ex vivo* (n = 19) and (**b**) after 12-day *in vitro* expansion (n = 9), and (**c**) in study participants of part I (n = 11) and part II (n = 24) after 12-day *in vitro* expansion of PBMCs collected prior to vaccination (day 1)

or on day 28 following vaccine administration. Functionality of CD4⁺ T cells was assessed for upregulation of the degranulation marker CD107a and production of the T helper 1 (Th1) cytokines (IFNγ, TNF, and IL 2). (**a–c**) Box plots or combined box-line plots display median with 25th or 75th percentiles, and min/max whiskers, two-sided Wilcoxon signed-rank test, n.a., not applicable; no, number; pos, positive.

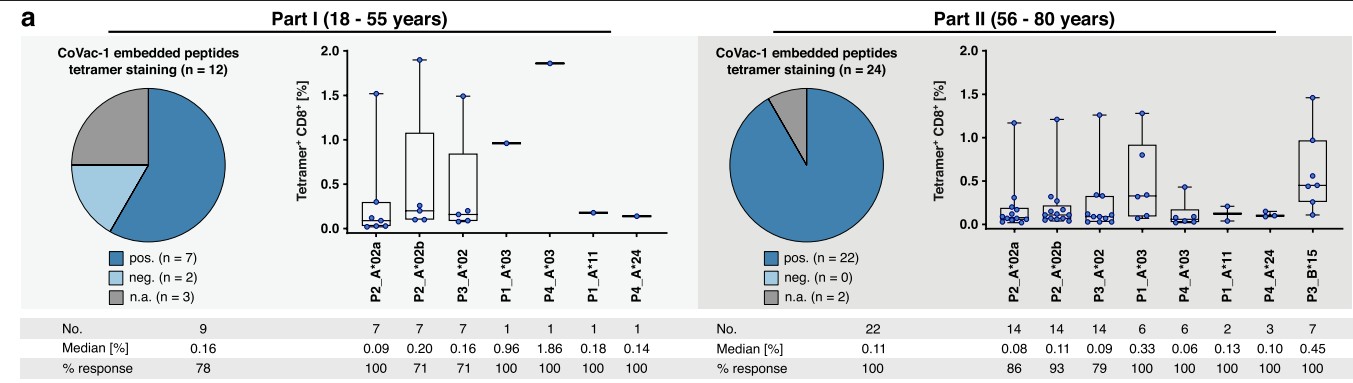

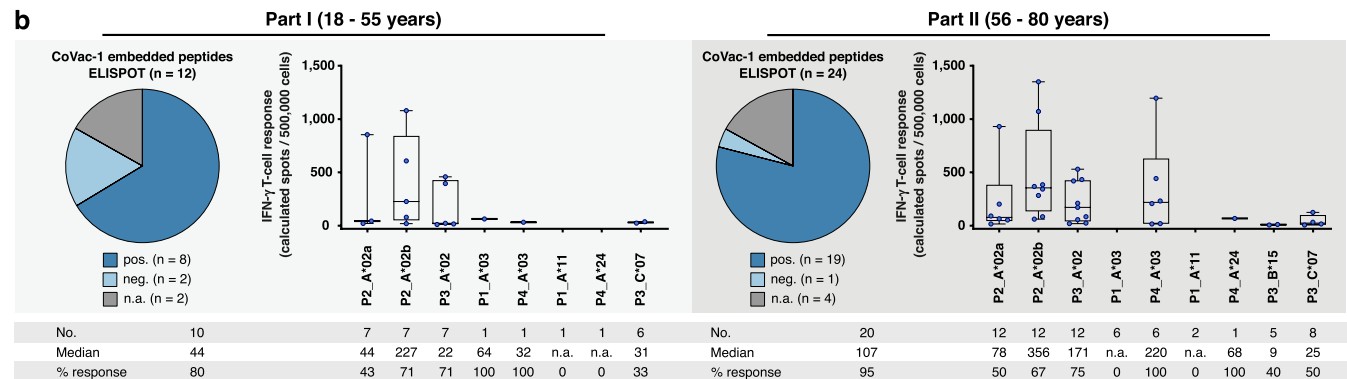

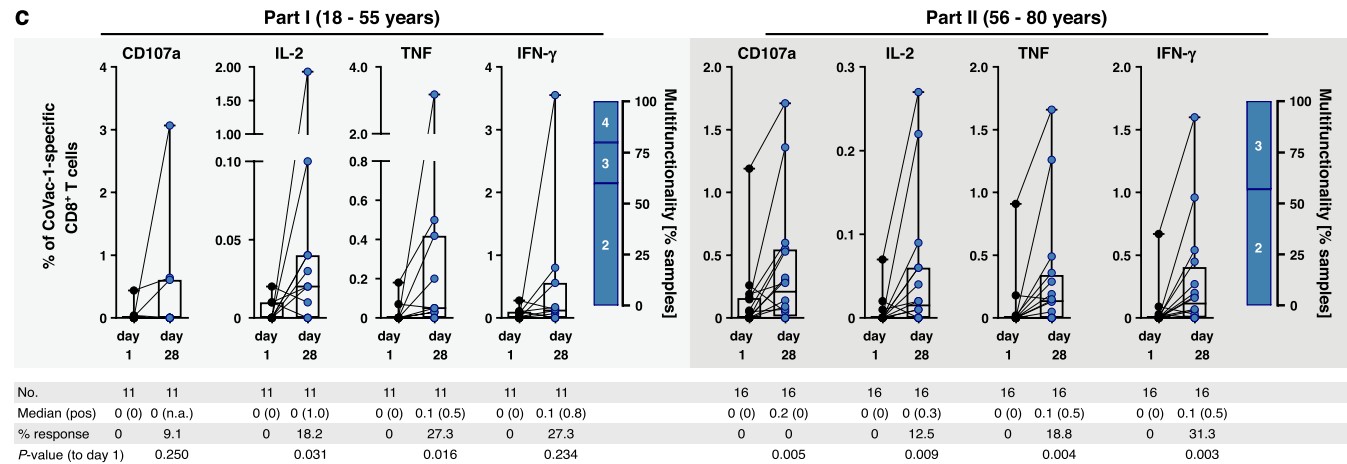

**Extended Data Fig. 6 | CoVac-1-induced CD8⁺ T cell responses to HLA class I-restricted CoVac-1-embedded peptides and CoVac-1 peptides.** T cell responses to HLA class I-restricted SARS-CoV-2 peptides embedded within the CoVac-1 vaccine peptides (matching the HLA allotype of the respective participant) were assessed by (**a**) tetramer staining and (**b**) IFNγ ELISPOT assays after *in vitro* expansion of PBMCs from study participants (part I and II) obtained on day 28 after vaccination. Pie charts display number of samples with (**a**) specific T cells or (**b**) IFNγ T cell responses to CoVac-1-embedded peptides (pos, positive; neg, negative; n.a., not assessed). Dots represent frequencies of peptide-specific T cells shown for individual donors with detected T cell responses only. (**c**) Frequencies of functional CoVac-1-induced CD4⁺ T cells in study participants prior to vaccination (day 1) and at day 28 following vaccination using intracellular cytokine (IFNγ, TNF, and IL-2) and surface marker staining (CD107a). The right graph displays the proportion of samples revealing difunctional (2), trifunctional (3), or tetrafunctional (4) T cell responses. (**a**–**c**) Box plots or combined box-line plots show median with 25th or 75th percentiles, and min/max whiskers, two-sided Wilcoxon signed-rank test. no, number; pos, positive.

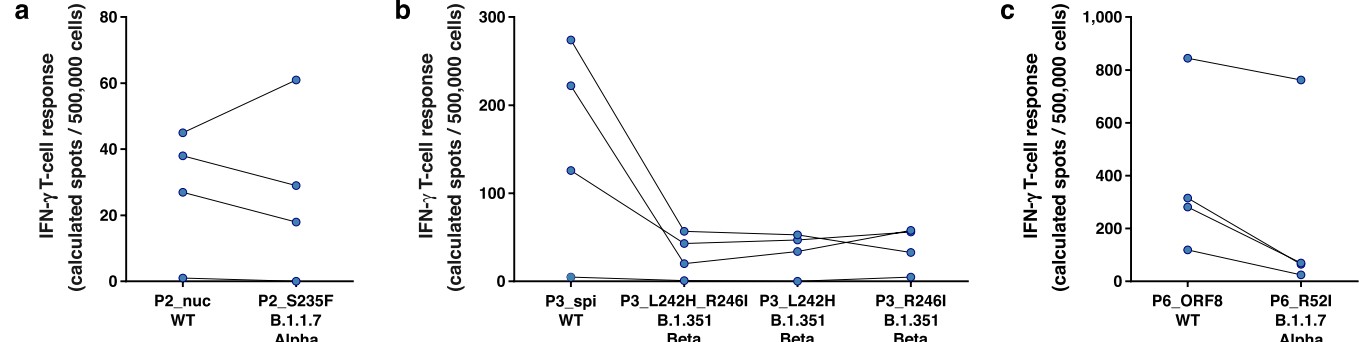

**Extended Data Fig. 7 | Vaccine-induced IFNγ T cell response to CoVac-1 peptides affected by mutations of SARS-CoV-2 variants of concern (VOC).** CoVac-1-induced T cell response to the single wild-type (WT) CoVac-1 peptides (P2_nuc (nucleocapsid), P3_spi (spike), P6_ORF8 (open reading frame 8)) in comparison to corresponding peptides comprising mutations of B.1.1.7-Alpha and B.1.351-Beta VOC were assessed by *ex vivo* IFNγ ELISPOT assay for (**a**) P2_nuc, (**b**) P3_spi, and (**c**) P6_ORF8 using PBMCs from study participants (n = 4) collected on day 28 after CoVac-1 vaccination.

**Extended Data Table 1 | Local and systemic solicited AEs compared between part I and II**

| Local AEs | Severity | All participants (n = 36) | Part I (n = 12) | Part II (n = 24) | P-value* |
|---|---|---|---|---|---|
| Any local event, n (%) | Mild | 14 (38.9) | 4 (33.3) | 10 (41.7) | |
| | Moderate | 15 (41.7) | 5 (41.7) | 10 (41.7) | 0.899 |
| | Severe | 7 (19.4) | 3 (25) | 4 (16.7) | |
| Induration, n (%) | Mild | 24 (66.7) | 8 (66.7) | 16 (66.7) | 1.000 |
| | Moderate | 12 (33.3) | 4 (33.3) | 8 (33.3) | |
| Swelling, n (%) | Mild | 20 (55.6) | 9 (75) | 11 (45.8) | |
| | Moderate | 7 (19.4) | 2 (16.7) | 5 (20.8) | 0.802 |
| | Severe | 2 (5.6) | 1 (8.3) | 1 (4.2) | |
| Erythema, n (%) | Mild | 16 (44.4) | 5 (41.7) | 11 (45.8) | |
| | Moderate | 10 (27.8) | 4 (33.3) | 6 (25) | 0.806 |
| | Severe | 7 (19.4) | 3 (25) | 4 (16.7) | |
| Itching, n (%) | Mild | 24 (66.7) | 9 (75) | 15 (62.5) | 0.709 |
| | Moderate | - | - | - | |
| Pain, n (%) | Mild | 21 (58.3) | 8 (66.7) | 13 (54.2) | 0.721 |
| | Moderate | 1 (2.8) | - | 1 (4.2) | |
| Skin ulceration, n (%) | Mild | 7 (19.4) | 3 (25) | 4 (16.7) | 0.664 |
| | Moderate | 2 (5.6) | - | 2 (8.3) | |
| Lymphadenopathy, n (%) | Mild | 8 (22.2) | 5 (41.7) | 3 (12.5) | 0.086 |
| | Moderate | - | - | - | |

| Systemic AEs | Severity | All participants (n = 36) | Part I (n = 12) | Part II (n = 24) | P-value* |
|---|---|---|---|---|---|
| Any systemic event, n (%) | Mild | 14 (38.9) | 5 (41.7) | 9 (37.5) | 1.000 |
| | Moderate | - | - | - | |
| Fatigue, n (%) | Mild | 11 (30.6) | 3 (25) | 8 (33.3) | 0.715 |
| | Moderate | - | - | - | |
| Headache, n (%) | Mild | 6 (16.7) | 2 (16.7) | 4 (16.7) | 1.000 |
| | Moderate | - | - | - | |
| Chills, n (%) | Mild | 2 (5.6) | - | 2 (8.3) | 0.543 |
| | Moderate | - | - | - | |
| Myalgia, n (%) | Mild | 2 (5.6) | 1 (8.3) | 1 (4.2) | 1.000 |
| | Moderate | - | - | - | |
| Arthralgia, n (%) | Mild | 1 (2.8) | 1 (8.3) | - | 0.333 |
| | Moderate | - | - | - | |
| Fever, n (%) | Mild | - | - | - | - |
| | Moderate | - | - | - | |

Related local and systemic solicited adverse events (AEs) assessed up to 56 days after vaccination. Severity was graded as mild (grade 1), moderate (grade 2), or severe (grade 3) based on the definition provided in the methods section. * P-values were calculated for the comparison of Part I and Part II using two-sided Fisher's Exact test. n, number.

**Extended Data Table 2 | Unsolicited AEs classified according to CTCAE V5.0**

| CTCAE | Severity | All participants (n = 36) | | Part I (n = 12) | | Part II (n = 24) | |
|---|---|---|---|---|---|---|---|
| | | Not related to vaccine | Related to vaccine | Not related to vaccine | Related to vaccine | Not related to vaccine | Related to vaccine |
| Any event | Mild | 46 | 1 | 17 | - | 29 | 1 |
| | Moderate | 9 | 1 | 4 | - | 5 | 1 |
| | Severe | 1 | - | 1 | - | - | - |
| Fatigue | Mild | 7 | - | 1 | - | 6 | - |
| | Moderate | - | - | - | - | - | - |
| Headache | Mild | 11 | - | 4 | - | 7 | - |
| | Moderate | - | - | - | - | - | - |
| Dysesthesia | Mild | 1 | - | - | - | 1 | - |
| | Moderate | - | - | - | - | - | - |
| Paresthesia | Mild | 1 | - | - | - | 1 | - |
| | Moderate | - | - | - | - | - | - |
| Nausea | Mild | 5 | - | 3 | - | 2 | - |
| | Moderate | - | - | - | - | - | - |
| Muscle cramp | Mild | 2 | - | 2 | - | - | - |
| | Moderate | - | - | - | - | - | - |
| Sore throat | Mild | 2 | - | 1 | - | 1 | - |
| | Moderate | - | - | - | - | - | - |
| Dizziness | Mild | 2 | - | 1 | - | 1 | - |
| | Moderate | - | - | - | - | - | - |
| Ear pain | Mild | 1 | - | 1 | - | - | - |
| | Moderate | - | - | - | - | - | - |
| Sinusitis | Mild | - | - | - | - | - | - |
| | Moderate | 1 | - | 1 | - | - | - |
| Diarrhea | Mild | 2 | - | 1 | - | 1 | - |
| | Moderate | - | - | - | - | - | - |
| Bloating | Mild | 1 | - | 1 | - | - | - |
| | Moderate | - | - | - | - | - | - |
| Pain in extremity | Mild | 2 | - | - | - | 2 | - |
| | Moderate | - | - | - | - | - | - |
| Aphtae oral | Mild | 1 | - | - | - | 1 | - |
| | Moderate | - | - | - | - | - | - |
| Hypertension | Mild | 2 | - | 2 | - | - | - |
| | Moderate | 3 | - | 3 | - | - | - |
| | Severe | 1 | - | 1 | - | - | - |
| Herpes simplex reactivation | Mild | - | 1 | - | - | - | 1 |
| | Moderate | - | - | - | - | - | - |
| Laceration left hand | Mild | - | - | - | - | - | - |
| | Moderate | 1 | - | - | - | 1 | - |
| Back pain | Mild | 1 | - | - | - | 1 | - |
| | Moderate | - | - | - | - | - | - |
| Retinopathy | Mild | - | - | - | - | - | - |
| | Moderate | 2 | - | - | - | 2 | - |
| Hot flashes | Mild | 1 | - | - | - | 1 | - |
| | Moderate | - | - | - | - | - | - |
| Joint effusion | Mild | - | - | - | - | - | - |
| | Moderate | 1 | - | - | - | 1 | - |
| Abdominal pain | Mild | 2 | - | - | - | 2 | - |
| | Moderate | - | - | - | - | - | - |
| Renal colic | Mild | 1 | - | - | - | 1 | - |
| | Moderate | - | - | - | - | - | - |
| Shingles | Mild | - | - | - | - | - | - |
| | Moderate | - | 1 | - | - | - | 1 |
| Allergic reaction | Mild | 1 | - | - | - | 1 | - |
| | Moderate | - | - | - | - | - | - |
| Suspicious skin lesion | Mild | - | - | - | - | - | - |
| | Moderate | 1 | - | - | - | 1 | - |

Severity and relationship were judged by the investigator until day 56. AE, adverse event; CTCAE, common terminology criteria for adverse events; n, number.

**Extended Data Table 3 | Comparison of immunogenicity between part I and part II**

| Characteristics of T-cell response | Part | Mean | SD | *P*-value* |
|---|---|---|---|---|
| IFN-γ ELISPOT day 7 - *ex vivo* [calculated spot count] - (Fig. 2a) | I | 8.33 | 10.12 | 0.123 |
| | II | 4.83 | 7.20 | |
| IFN-γ ELISPOT day 14 - *ex vivo* [calculated spot count] - (Fig. 2a) | I | 454.50 | 665.36 | 0.084 |
| | II | 234.38 | 373.25 | |
| IFN-γ ELISPOT day 28 - *ex vivo* [calculated spot count] - (Fig. 2a) | I | 887.42 | 1,183 | 0.481 |
| | II | 657.08 | 795.38 | |
| Induced CoVac-1 peptides day 7 [n] - (Fig. 2b) | I | 0.08 | 0.29 | 0.612 |
| | II | 0.04 | 0.20 | |
| Induced CoVac-1 peptides day 14 [n] - (Fig. 2b) | I | 4.50 | 1.38 | 0.110 |
| | II | 3.21 | 2.19 | |
| Induced CoVac-1 peptides day 28 [n] - (Fig. 2b) | I | 5.08 | 0.67 | 0.901 |
| | II | 4.88 | 1.36 | |
| CD107a+ CD4+ T cells day 28 - *ex vivo* [%] - (Fig. 2d) | I | 0.18 | 0.12 | 0.253 |
| | II | 0.18 | 0.21 | |
| IL-2+ CD4+ T cells day 28 - *ex vivo* [%] - (Fig. 2d) | I | 0.82 | 0.87 | 0.907 |
| | II | 1.12 | 1.95 | |
| TNF+ CD4+ T cells day 28 - *ex vivo* [%] - (Fig. 2d) | I | 0.93 | 0.92 | 1.000 |
| | II | 1.23 | 2.14 | |
| IFN-γ+ CD4+ T cells day 28 - *ex vivo* [%] - (Fig. 2d) | I | 0.53 | 0.58 | 0.724 |
| | II | 0.59 | 0.82 | |
| IFN-γ ELISPOT day 56 - *ex vivo* [calculated spot count] - (Extended Data Fig. 3a) | I | 321.75 | 303.82 | 0.639 |
| | II | 501.63 | 545.8 | |
| IFN-γ ELISPOT month 3 - *ex vivo* [calculated spot count] - (Extended Data Fig. 3a) | I | 192.31 | 185.71 | 0.365 |
| | II | 398.09 | 468.93 | |
| IFN-γ ELISPOT day 28 - 12d IVE [calculated spot count] - (Extended Data Fig. 3a) | I | 10,299 | 4,214 | 0.546 |
| | II | 9,440 | 5,137 | |
| IFN-γ ELISPOT day 56 - 12d IVE [calculated spot count] - (Extended Data Fig. 3a) | I | 12,049 | 7,093 | 0.019 |
| | II | 6,985 | 3,075 | |
| IFN-γ ELISPOT month 3 - 12d IVE [calculated spot count] - (Extended Data Fig. 3a) | I | 11,650 | 5,109 | 0.032 |
| | II | 8,521 | 4,425 | |
| CD107a+ CD4+ T cells day 28 - 12d IVE [%] - (Extended Data Fig. 5c) | I | 6.59 | 4.02 | 0.131 |
| | II | 11.33 | 8.65 | |
| IL-2+ CD4+ T cells day 28 - 12d IVE [%] - (Extended Data Fig. 5c) | I | 3.82 | 1.51 | 0.007 |
| | II | 10.92 | 7.85 | |
| TNF+ CD4+ T cells day 28 - 12d IVE [%] - (Extended Data Fig. 5c) | I | 13.67 | 7.44 | 0.136 |
| | II | 20.82 | 13.28 | |
| IFN-γ+ CD4+ T cells day 28 - 12d IVE [%] - (Extended Data Fig. 5c) | I | 7.18 | 3.37 | 0.065 |
| | II | 12.06 | 8.47 | |
| CD107a+ CD8+ T cells day 28 - 12d IVE [%] - (Extended Data Fig. 6c) | I | 0.39 | 0.92 | 0.148 |
| | II | 0.38 | 0.50 | |
| IL-2+ CD8+ T cells day 28 - 12d IVE [%] - (Extended Data Fig. 6c) | I | 0.20 | 0.57 | 0.724 |
| | II | 0.05 | 0.08 | |
| TNF+ CD8+ T cells day 28 - 12d IVE [%] - (Extended Data Fig. 6c) | I | 0.41 | 0.94 | 1.000 |
| | II | 0.30 | 0.48 | |
| IFN-γ+ CD8+ T cells day 28 - 12d IVE [%] - (Extended Data Fig. 6c) | I | 0.49 | 1.05 | 0.881 |
| | II | 0.28 | 0.44 | |

* *P*-values were calculated for the comparison of Part I and Part II using two-sided Mann-Whitney U-test. 12d IVE, 12-day *in vitro* expansion; n, number; SD, standard deviation.

# Reporting Summary

## Statistics

For all statistical analyses, confirm that the following items are present in the figure legend, table legend, main text, or Methods section.

| n/a | Confirmed | |
|---|---|---|
| ☐ | ☒ | The exact sample size (*n*) for each experimental group/condition, given as a discrete number and unit of measurement |
| ☐ | ☒ | A statement on whether measurements were taken from distinct samples or whether the same sample was measured repeatedly |
| ☐ | ☒ | The statistical test(s) used AND whether they are one- or two-sided *Only common tests should be described solely by name; describe more complex techniques in the Methods section.* |
| ☐ | ☒ | A description of all covariates tested |
| ☐ | ☒ | A description of any assumptions or corrections, such as tests of normality and adjustment for multiple comparisons |
| ☐ | ☒ | A full description of the statistical parameters including central tendency (e.g. means) or other basic estimates (e.g. regression coefficient) AND variation (e.g. standard deviation) or associated estimates of uncertainty (e.g. confidence intervals) |
| ☐ | ☒ | For null hypothesis testing, the test statistic (e.g. $F$, $t$, $r$) with confidence intervals, effect sizes, degrees of freedom and $P$ value noted *Give P values as exact values whenever suitable.* |
| ☒ | ☐ | For Bayesian analysis, information on the choice of priors and Markov chain Monte Carlo settings |
| ☒ | ☐ | For hierarchical and complex designs, identification of the appropriate level for tests and full reporting of outcomes |
| ☒ | ☐ | Estimates of effect sizes (e.g. Cohen's *d*, Pearson's *r*), indicating how they were calculated |

*Our web collection on statistics for biologists contains articles on many of the points above.*

## Software and code

Policy information about availability of computer code

| Data collection | Patient diary for the first 28 days after vaccination and regular visits at the trial site. Data on reactogenicity and immunogenicity were collected in an electronic case report form. Additional data on explorative endpoints was provided via electronic sheets. |
|---|---|
| Data analysis | GraphPad Prism 9.2.0, FlowJo software version 10.7.1 and SAS 9.4. |

For manuscripts utilizing custom algorithms or software that are central to the research but not yet described in published literature, software must be made available to editors and reviewers. We strongly encourage code deposition in a community repository (e.g. GitHub). See the Nature Portfolio guidelines for submitting code & software for further information.

## Data

Policy information about availability of data

All manuscripts must include a data availability statement. This statement should provide the following information, where applicable:
- Accession codes, unique identifiers, or web links for publicly available datasets
- A description of any restrictions on data availability
- For clinical datasets or third party data, please ensure that the statement adheres to our policy

Data supporting the findings of this study including the study protocol and the statistical analysis plan are supplied as source data with this manuscript. Further data, including de-identified participant data are available after final completion of the trial report and are shared according to data sharing guidelines upon reasonable request to the corresponding author.

# Field-specific reporting

Please select the one below that is the best fit for your research. If you are not sure, read the appropriate sections before making your selection.

☒ Life sciences ☐ Behavioural & social sciences ☐ Ecological, evolutionary & environmental sciences

For a reference copy of the document with all sections, see nature.com/documents/nr-reporting-summary-flat.pdf

# Life sciences study design

All studies must disclose on these points even when the disclosure is negative.

| | |
|---|---|
| Sample size | The sample size calculation of 36 participants for the trial was based on the stopping rule (occurrence of a vaccine-related serious adverse event (SAE)) and determined to show that the incidence of SAE associated with administration of CoVac-1 does not exceed a predetermined rate of 5% (= P1). Safety of the CoVac-1 vaccine should be shown if no SAE (= P0) occurs in the study population. An evaluable sample size of 33 achieves 81.6% of power to detect a difference (P1 - P0) of 0.0499 using a one-sided exact test based on the binomial distribution with a target significance level of 0.05 (sample size determination using PASS). These results assume that the population proportion under the null hypothesis (P0) is ≤ 0.0001. Assuming a drop-out rate of 7.5% (percentage of subjects that are expected to be lost at random during the course of the study and for whom no response data concerning existence of SAE will be collected, i.e. will be treated as "missing") the total number of 36 subjects should be enrolled in the study to achieve the threshold of 33 evaluable subjects. In total 36 subjects were included in the study. |
| Data exclusions | Safety and immunogenicity data were available until day 56 and month 3 after vaccination, respectively. No data were excluded from the analyses. Samples analyzed are indicated in the respective figure caption. |
| Replication | This is a report of an ongoing clinical trial. So far no attempt to replicate was performed. |
| Randomization | There was no randomization in this clinical trial as there was only one treatment arm without a control arm. |
| Blinding | There was no blinding performed in this clinical trial, because all participants received the CoVac-1 vaccine. |

# Reporting for specific materials, systems and methods

We require information from authors about some types of materials, experimental systems and methods used in many studies. Here, indicate whether each material, system or method listed is relevant to your study. If you are not sure if a list item applies to your research, read the appropriate section before selecting a response.

## Materials & experimental systems

| n/a | Involved in the study |
|---|---|
| ☐ | ☒ Antibodies |
| ☒ | ☐ Eukaryotic cell lines |
| ☒ | ☐ Palaeontology and archaeology |
| ☒ | ☐ Animals and other organisms |
| ☐ | ☒ Human research participants |
| ☐ | ☒ Clinical data |
| ☒ | ☐ Dual use research of concern |

## Methods

| n/a | Involved in the study |
|---|---|
| ☒ | ☐ ChIP-seq |
| ☐ | ☒ Flow cytometry |
| ☒ | ☐ MRI-based neuroimaging |

# Antibodies

| | |
|---|---|
| Antibodies used | Flow cytometry: APC/Cy7 anti-human CD4 (clone RPA-T4, Cat# 300518, BioLegend), PE/Cy7 anti-human CD8 (clone SFCI21Thy2D3, Cat# 737661, Beckman Coulter), Pacific Blue anti-human TNF (clone MAb11, Cat# 502920, BioLegend), FITC anti-human CD107a (clone H4A3, Cat# 328606, BioLegend), APC anti-human IL-2 (clone MQ1-17H12, Cat# 500309, BioLegend), and PE anti-human IFN-γ monoclonal antibodies (clone B27, Cat# 506507, BioLegend). ELISPOT: anti-IFN-γ antibody (clone 1-D1K, Cat# 3420-3-1000, MabTech), anti-IFN-γ biotinylated detection antibody (clone 7-B6-1, Cat# 3420-6-1000 MabTech). |
| Validation | All antibodies were purchased from the above stated companies. Validation data / citations can be found on the suppliers' website. APC/Cy7 anti-human CD4: https://www.biolegend.com/en-us/products/apc-cyanine7-anti-human-cd4-antibody-1933 PE/Cy7 anti-human CD8: https://www.beckman.de/search#q=737661&t=coveo-tab-techdocs&f:@category=[Certificates%20of%20Analysis]&f:@itemnumber=[737661] Pacific Blue anti-human TNF: https://www.biolegend.com/en-us/products/pacific-blue-anti-human-tnf-alpha-antibody-4149 FITC anti-human CD107a: https://www.biolegend.com/en-us/products/fitc-anti-human-cd107a-lamp-1-antibody-4966 APC anti-human IL-2: https://www.biolegend.com/en-us/products/apc-anti-human-il-2-antibody-1348 PE anti-human IFN-γ: https://www.biolegend.com/en-us/products/pe-anti-human-ifn-gamma-antibody-1536 anti-IFN-γ antibody: https://www.mabtech.com/products/anti-human-ifn-gamma-antibody-1-d1k-purified-3420-3 |

anti-IFN-γ biotinylated detection antibody: https://www.mabtech.com/products/anti-human-ifn-gamma-antibody-7-b6-1-biotinylated-3420-6#tabs-min-2

# Human research participants

Policy information about studies involving human research participants

| | |
|---|---|
| Population characteristics | Eligible participants were men or women aged 18-55 (Part I) or 56-80 years (Part II). In Part I, participants were free of clinically significant health problems. In Part II, participants with stable medical history were enrolled. Participants had to refrain from blood donations during the course of the study and be willing to minimize body fluid transmission to others for 7 days after vaccination. All participants had to adhere to adequate contraception methods until three months after vaccination. |
| | Exclusion criteria comprised: pregnant or lactating females, participation in another clinical trial, treatment with immunosuppressive drugs, prior or current infection with SARS-CoV-2 (proven serologically or by PCR), known previous anaphylactic reaction to any component or hypersensitivity to any component of the CoVac-1 vaccine, relevant CNS pathology or other neurological disease, positivity for HIV or active hepatitis, lymphocyte count ≤1.000/µl, blood donation within 30 days, or administration of immunoglobulins or blood products within 120 days prior to study inclusion, diabetes type II, chronic lung disease requiring drug treatment, increased liver enzymes (≥2.5 x upper limit of normal), renal failure (GFR<60ml/min/1.73m2), serious cardiovascular disease, sickle cell anemia, obesity (defined by age adjusted body mass index), or preexisting auto-immune disease except for Hashimoto thyroiditis and mild psoriasis. |
| Recruitment | Participants were recruited at the University Hospital Tübingen. Information on the clinical trial was provided via press release (electronic and paper based). A selection bias is not assumed. Recruited participants were screened for eligibility. First the Part I of the trial was completed (including sentinel dosing of the first participant) and after review of reactogenicity and immunogenicity by the data safety monitoring board and approval by the regulatory authorities (Paul Ehrlich Institute and local ethic committee), Part II of the trial was initiated. |
| Ethics oversight | The trial was approved by the Ethics Committee, University Tübingen (537/2020AMG1) and the Paul Ehrlich Institute and performed in accordance with the International Council for Harmonization Good Clinical Practice guideline. A second approval was obtained prior to recruiting in Part II of the clinical trial. |

Note that full information on the approval of the study protocol must also be provided in the manuscript.

# Clinical data

Policy information about clinical studies
All manuscripts should comply with the ICMJE guidelines for publication of clinical research and a completed CONSORT checklist must be included with all submissions.

| | |
|---|---|
| Clinical trial registration | ClinicalTrials.gov: NCT04546841 |
| Study protocol | The study protocol is provided with the submission of the manuscript. |
| Data collection | Participants were recruited from November 28th, 2020 to January 15th, 2021. Data were collected at screening (up to 7 days before vaccination), day 1 (vaccination, baseline), day 7, day 14, day 28, day 56 and month 3. Both reactogenicity and immunogenicity were analyzed at indicated time points by outpatients visits at the University Hospital Tübingen. In addition, participants reported on reactogenicity until day 28 by paper-based diary. |
| Outcomes | In this report, safety as the primary endpoint is presented until day 56. Primary safety outcomes reflect the nature, frequency, and severity of solicited adverse events (AEs) until day 56 after vaccination. In addition, the number and percentage of participants with unsolicited events until day 56 were reported. |
| | The secondary endpoint immunogenicity is reported by the induction of CoVac-1-specific T-cell responses. Furthermore, explorative endpoints such as characteristics of T-cell responses were analyzed. |

# Flow Cytometry

## Plots

Confirm that:

☒ The axis labels state the marker and fluorochrome used (e.g. CD4-FITC).

☒ The axis scales are clearly visible. Include numbers along axes only for bottom left plot of group (a 'group' is an analysis of identical markers).

☒ All plots are contour plots with outliers or pseudocolor plots.

☒ A numerical value for number of cells or percentage (with statistics) is provided.

## Methodology

| | |
|---|---|
| Sample preparation | PBMCs were incubated with 10 µg/mL of peptide, 10 µg/mL Brefeldin A (Sigma-Aldrich), and a 1:500 dilution of GolgiStop (BD) for 12 - 16 h. Staining was performed using Cytofix/Cytoperm solution (BD), APC/Cy7 anti-human CD4 (BioLegend), PE/Cy7 anti-human CD8 (Beckman Coulter), Pacific Blue anti-human TNF, FITC anti-human CD107a, APC anti-human IL-2, and PE |

anti-human IFN-γ monoclonal antibodies (BioLegend). PMA (5 μg/mL) and ionomycin (1 μM, Sigma-Aldrich) served as positive control. Viable cells were determined using Aqua live/dead (Invitrogen).

| | |
|---|---|
| Instrument | FACS Canto II cytometer (BD) |
| Software | FlowJo software version 10.7.1 (BD) |
| Cell population abundance | No cell sorting was performed prior to functional experiments. |
| Gating strategy | Viable cells were determined using Aqua live/dead (Invitrogen). |

☒ Tick this box to confirm that a figure exemplifying the gating strategy is provided in the Supplementary Information.

