## [Peer Review File · Nature]

Manuscript Title: A COVID-19 Peptide Vaccine for the Induction of SARS-CoV-2 T-Cell Immunity

Reviewer Comments & Author Rebuttals

Reviewer Reports on the Initial Version:

Referee #1 (Remarks to the Author):

This manuscript describes the safety and immunogenicity of a new peptide-based SARS-CoV-2 vaccine designed to induce SARS-CoV-2 specific T cells (CD4 and CD8) against different SARS-CoV-2 proteins. The vaccine is composed of 6 different peptides covering degenerate HLA-DR peptide from NP, Spike, env, membrane and ORF8 with embedded HLA-class I peptides. It should therefore stimulate T cell response (CD4 and CD8) in a population carrying different HLA-Class I and class II molecules.

The authors performed this analysis in two groups of individuals (12 individuals age 18-55, 24 individuals age 56-80). Safety was analyzed until day 56. T cell response at day 7, 14 and 28.

They reported good safety record and the induction of a robust and multispecific (at least 5-6 epitopes recognized x vaccinated individual) T cell response in 100% of vaccinated individuals. The authors also claimed that vaccine-induced T cells is phenotypically similar to what is induced by natural infection and that also quantitatively more robust to SARS-CoV-2 T cells present in COVID-19 convalescent. They also reported that vaccine-induced T cells are marginally affected by mutations present in current VOCs. Interestingly they can also detect CD8 T cells at least in HLA-A0201, HLA-B15 and HLA-A03+ individuals.

The induction of a pure SARS-CoV-2 T cell vaccine might be, as the authors discussed, of importance for selected individuals with B cell deficiency.

Overall, the work is interesting, novel and well conducted. The analysis of T cells performed at high standard and in general the results support the conclusions. It is certainly of interest to show that a peptide vaccine composed of only 6 peptides can elicit a T cell response in 100% of vaccinated individuals. There are however some limitations.

Major Comments

- a) One limitation of the present manuscript is that, as the authors underlined, the scope of a T-cell inducing vaccine is to induce a long term memory immunity. Clearly an analysis of T cell response performed only until 28 days after vaccination cannot answer this question. As such it will be important to follow up the level of SARS-CoV-2 T cells induced by the peptide-based vaccine at least at day 60-90 and see whether T cells frequency is maintained as such high level for longer time.
- b) The comparison between the magnitude of T cell induced by the peptide-based vaccines and the natural infection should be done at equivalent time after induction. It is not completely correct to compare frequency of T cells induced by vaccine at day 28 with general number of SARS-CoV-2 T cells detected in convalescent at not defined time after infection. Even though the frequency of SARS-CoV-2 T cell induced by this peptide-based vaccine is indeed very high, a precise comparison should be done with samples taken at equivalent time after induction.
- c) Similarly, I think it will be important to compare the overall magnitude of SARS-CoV-2 T cells induced by the peptide-based vaccine with the quantity of Spike-specific T cells induced by current mRNA or Adeno-based vaccines. Such analysis could be very informative to evaluate the potential advantage of this new vaccine with the ones currently available.
- d) T cell affinity measurement can be one other additional information that can help to better

define the T cells induced by this novel peptide-based vaccine. I cannot find any indication in the material and methods of the concentration of peptides used to stimulate PBMC in the different assays. The authors should add this information. In addition, it will be of interest to show that the T cells induced by this peptide-vaccine are not low affinity T cells. An analysis of the functional affinity of the SARS-CoV-2 T cells induced by the peptide-based vaccine should be added. A simple experiment performed with different concentration of peptides in few selected vaccinated can indicate whether T cells recognize high or low quantity of peptides. A comparison with SARS-CoV-2 T cells induced by natural infection can also complete the characterization of the vaccine-primed T cells.

Referee #2 (Remarks to the Author):

Title: Phase I Trial of a Multi-Peptide COVID-19 Vaccine for the Induction of SARS-CoV2 T-cell Immunity

A. Summary of the key results

The manuscript by Heitmann et.al reports the results of phase I clinical trial aimed to evaluate safety, tolerability, and immunogenicity of CoVac-1, a novel multi-peptide COVID-19 vaccine, administered with the toll-like receptor (TLR)1/2 agonist XS15 emulsified in Montanide™ ISA51 VG to healthy adults ages 18-80 as a subcutaneous single injection. Overall, CoVac-1 vaccination demonstrated a favorable safety profile. Solicited local reactogenicity was mild to moderate in severity with several instances of severe erythema and swelling. All study participants developed granuloma at the injection site, which was reported as an expected outcome due to the use of the novel TLR1/2 XS15 adjuvant. Solicited systemic reactogenicity was absent or mild. CoVac-1 vaccine was shown to be highly immunogenic and induced polyfunctional CoVac-1 specific CD4+ and CD8+ T cells with high levels of expression of IFN- γ , TNF- α , and IL-2 28 days after vaccination. Notably, the authors claimed that SARS-CoV2 variants of concern (VOC) do not impact vaccine immunogenicity.

B. Originality and significance: if not novel, please include reference

One of the main strengths of the manuscript is that it addresses the important and timely subject of designing a vaccine that can potentially induce rapid, broad-spectrum, and long-lasting immunity against COVID-19, even in populations with B cell deficiencies, both congenital and acquired, as well as the elderly. The authors designed an innovative peptide-based vaccine utilizing immunotherapy principles. Unlike many SARS-CoV2 vaccines that almost solely rely on SARS-CoV2 Spike glycoprotein (S) to induce immune response, CoVac-1 vaccine incorporates 6 dominant T cell epitopes derived from viral proteins that are associated with COVID-19 immunity. This is an interesting approach that could lead to advanced protection against the infection. This trial also incorporates the use of a novel, previously untested in clinical trials, adjuvant TLR1/2 ligand XS15, which was specifically designed to induced CD4+ and CD8+ immune responses. Furthermore, the results of this trial suggest that this vaccination strategy could potentially provide optimal protection, regardless of emerging VOC. However, the durability of immune responses was shown up to day 28 only, and therefore it is unclear whether this vaccine offers long term protection or if there is a chance of breakthrough infection.

C. Data & methodology: validity of approach, quality of data, quality of presentation

In terms of trial design, dose rationale for either peptides or an adjuvant was not clearly described in the manuscript and therefore makes it difficult for the reader to understand why this trial was not a dose escalation study with several doses being evaluated or at what lowest dose the immunogenicity could be achieved. Authors are encouraged to revise the methodology section to provide clarity. No data or description of methodology or evaluation of the "sentinel" case was provided.

D. Appropriate use of statistics and treatment of uncertainties

The statistical analysis section presented in the manuscript merits some significant revisions. As written (line 285-290, S141-146), it appears that the only statistical analysis performed was the sample size calculation; no subsequent statistical analyses of any data sets were described. It is very important for the authors to use appropriate statistical methodologies to provide a complete representation of the data acquired in the trial. This would include proper statistical comparisons/test between the cohorts and between different time points, description of how fold changes were calculated, and providing p-values where relevant. Main and extended figures need to be revised to include relevant statistical comparisons. Also, please consider evaluating the differences between % of CoVac-1 specific CD4 vs CD8 T cells, and the differences seen in specific cytokine production between the cohorts.

E. Conclusions: robustness, validity, reliability

The authors seem to claim that CoVac-1 vaccination provides a "superior T cell immunity" (Line 190, 218), however this generalization is not fully supported by the data presented in the manuscript. Although, study results demonstrated that the vaccine induced polyfunctional T cell responses that are indeed correlates of protection against the disease, these responses are reported up to day 28 only. The authors also note the efficacy of the investigational candidate (line 74,269) but the study was not designed or powered to demonstrate efficacy. The authors are encouraged to revise the language to improve readability and avoid overstatements.

F. Suggested improvements: experiments, data for possible revision

Is it possible to develop antibodies against vaccine epitopes following immunization with CoVac-1? Was this evaluated? Could the authors comment on this?

Vaccination with CoVac-1 is intended to stimulate cellular immunity, however did the authors evaluate SARS-CoV2 specific antibody responses, were they induced by the vaccine?

Including immunological data beyond day 28 will strengthen the manuscript and will be a welcome addition if available.

Line 79-87: Please, include percentage of males and females in the study as well as mean (SD) age

Line 90: What was the duration of diary cards?

Line 93,95, 102: Was there any variation in reported reactogenicity between the age groups? How long did it take for grade 3 AE to resolve? Please, specify.

Lines 94-95, 97-98: How long did it take for granulomas/skin ulcerations to resolve?

Line 129: Briefly state rationale for a 12-day in vitro expansion protocol.

Line 141: Were there any vaccine induced CD8+ T cell response observed ex vivo or they were too low frequency to detect?

Please, consider adding a supplemental table to complement Figure 1 that would include actual percentages of subjects that developed each of the local and systemic reactogenicity events.

Figure 2B: in the legend, please specify what method was used to measure CoVac-1 specific peptides

Figure 2D: please revise y-axis label to clearly show that the cells were CoVac-1 specific CD4 T cells

Extended figure 4B demonstrates the data of 12-day in vitro expansion of CoVac-1 induced CD4 T cells, is there similar data available for human convalescent samples?

G. References: appropriate credit to previous work?

Yes, however, please, review journal specific guidelines to reduce number of references accordingly

H. Clarity and context: lucidity of abstract/summary, appropriateness of abstract, introduction and conclusions

Major: The authors are recommended to revise the structuring of the paper to better guide the

reader through introduction, methods, and results in advance of discussion. It is difficult to follow the logic of the trial design without being provided sufficient background information within the manuscript (e.g., the rationale for utilizing TLR ½ XS15 adjuvant and Montanide, selecting vaccine dose and regimen, peptide combination, CoVac-1 studies in pre-clinical models if available); while reading Line 59-63: the rationale for combining these 6 specific peptides does not seem immediately clear, please briefly discuss how recognizing multiple peptides could contribute to the development of a stronger immunity. Furthermore, is this the first report of XS15 adjuvant use in a human clinical trial? This also needs to be clarified.

Minor:

Lines 38, 76, 81, 227: although the trial intended to enroll participants up to 80 years of age, the upper limit was 70 based on the data presented in Table 1. This needs to be noted somewhere in the manuscript.

Line 115, 120, 152: what methods were used here?

Line 224: include clinicaltrials.gov identifier (NCT)

Line 245: please, specify the peptides used here

Author Rebuttals to Initial Comments:

Referee #1

This manuscript describes the safety and immunogenicity of a new peptide-based SARS-CoV-2 vaccine designed to induce SARS-CoV-2 specific T cells (CD4 and CD8) against different SARS-CoV-2 proteins. The vaccine is composed of 6 different peptides covering degenerate HLA-DR peptide from NP, Spike, env, membrane and ORF8 with embedded HLA-class I peptides. It should therefore stimulate T cell response (CD4 and CD8) in a population carrying different HLA-Class I and class II molecules.

The authors performed this analysis in two groups of individuals (12 individuals age 18-55, 24 individuals age 56-80. Safety was analyzed until day 56. T cell response at day 7,14 and 28.

They reported good safety record and the induction of a robust and multispecific (at least 5-6 epitopes recognized x vaccinated individual) T cell response in 100% of vaccinated individuals. The authors also claimed that vaccine-induced T cells is phenotypically similar to what is induced by natural infection and that also quantitatively more robust to SARS-CoV-2 T cells present in COVID-19 convalescent. They also reported that vaccine-induced T cells are marginally affected by mutations present in current VOCs. Interestingly they can also detected CD8 T cells at least in HLA-A0201, HLA-B15 and HLA-A03+ individuals.

The induction of a pure SARS-CoV-2 T cell vaccine might be, as the authors discussed, of importance for selected individuals with B cell deficiency.

Overall, the work is interesting, novel and well conducted. The analysis of T cells performed at high standard and in general the results support the conclusions. It is certainly of interest to show that a peptide vaccine composed of only 6 peptides can elicit a T cell response in 100% of vaccinated individuals. There are however some limitations.

Author reply: We thank the reviewer for the kind review of our manuscript and highly appreciate the input on how to further improve our work. Please find below a detailed description on how we addressed the raised points.

Major Comments

Comment 1: *One limitation of the present manuscript is that, as the authors underlined, the scope of a T-cell inducing vaccine is to induce a long-term memory immunity. Clearly an analysis of T cell response performed only until 28 days after vaccination cannot answer this question. As such it will be important to follow up the level of SARS-CoV-2 T cells induced by the peptide-based vaccine at least at day 60-90 and see whether T cells frequency is maintained as such high level for longer time.*

Author reply: We fully agree that follow-up assessment of CoVac-1-induced T-cell responses is of utmost importance to prove vaccine-induced long-term T-cell immunity. Therefore, the study protocol comprises follow-up sample collections and T-cell response analyses beyond day 28 after vaccination. To follow the reviewer's suggestion, which was likewise raised by Referee #2, we included follow-up data on vaccine-induced T-cell responses analyzed *ex vivo* and after 12-day *in vitro* expansion on day 56 as well as at month 3 post CoVac-1 vaccination in the revised manuscript (lines 147-152, Extended Data Fig. 3a). We could show that CoVac-1-induced T-cell responses persisted in the follow-up analyses at day 56 and month 3 after vaccination in all participants of Part I and Part II, with a decreasing IFN- γ T-cell response intensity observed *ex vivo* in Part I participants over time. However, profound and equivalent expandability of CoVac-1-induced T cells in both Part I and Part II participants was observed at month 3 compared to day 28 post vaccination, thereby indicating effective T-cell response upon virus challenge despite the expected decrease of *ex vivo* circulating CoVac-1-specific T cells over time (Dan *et al.*, Science, 2021; Bertoletti *et al.*, Cell Mol Immunol, 2021).

Comment 2: *The comparison between the magnitude of T cell induced by the peptide-based vaccines and the natural infection should be done at equivalent time after induction. It is not completely correct to compare frequency of T cells induced by vaccine at day 28 with general number of SARS-CoV-2 T cells detected in convalescent at not defined time after infection. Even though the frequency of SARS-CoV-2 T cell induced by this peptide-based vaccine is indeed very high, a precise comparison should be done with samples taken at equivalent time after induction.*

Author reply: We thank the reviewer for making this point. For the comparison of CoVac-1-induced T-cell responses on day 28 after vaccination with T-cell responses after natural SARS-CoV-2 infection the availability of samples taken exactly 28 days after the start of infection was very limited. Thus, we used

samples from COVID-19 human convalescents (HCs) collected 16-52 days (median 41-45 days within the three HC cohorts) after positive real-time polymerase chain reaction (PCR). A detailed information on the time points post infection for the different HCs cohorts can be found in Supplementary Table S2. We agree that the non-equivalent time points of sample collection could represent a limitation for the comparison of vaccine-induced and infection-induced T-cell responses. However, based on our previous work (Bilich *et al.*, Science Transl. Med. 2021) showing that the frequency of SARS-CoV-2 T-cell responses after natural infection remains robust up to six months after infection, it is suggested that the frequency of SARS-CoV-2 T-cell responses assessed between day 16 and day 52 post infection is not showing a large variation. To further address this issue, we included the new follow-up data on the frequencies of CoVac-1-induced T-cell responses on day 56 after vaccination in the revised Figure 2 to cover more equivalent the time points of sample collection after vaccination and natural infection in this comparative analysis. In addition, we described the limitation of non-equivalent time points of sample collection for the comparison of vaccine-induced and infection-induced T-cell responses in the revised Discussion section of the manuscript (lines 283-285).

Comment 3: *Similarly, I think it will be important to compare the overall magnitude of SARS-CoV-2 T cells induced by the peptide-based vaccine with the quantity of Spike-specific T cells induced by current mRNA or Adeno-based vaccines. Such analysis could be very informative to evaluate the potential advantage of this new vaccine with the ones currently available.*

Author reply: We fully agree with the reviewer that a comparison of CoVac-1-induced T-cell responses with Spike-specific T-cell responses induced by approved vaccines is of interest. To address this point, we analyzed a cohort of healthy volunteers (HVs), who received the mRNA vaccines BNT162b2 (n = 17) or mRNA-1273 (n = 3), an adenoviral vector-based vaccine (AZD1222, n = 5), or heterologous vaccination (AZD1222 vector-based vaccination followed by mRNA-1273 vaccine, n = 5). Using the PBMCs of these HVs collected 18-42 days after second vaccination, we performed *ex vivo* IFN- γ ELISPOT assays using overlapping 15-mer peptide pools covering the whole spike protein (Milteniy PepTivator® SARS-CoV-2 Prot_S, PepTivator® SARS-CoV-2 Prot_S+, PepTivator® SARS-CoV-2 Prot_S1). Detailed characteristics of the vaccinated volunteers are provided in a new Supplementary Table S3. We could detect vaccine-induced Spike-specific T-cell responses in 16/17 (94%), 4/5 (80%), and 5/5 (100%) of donors vaccinated with an mRNA-based, vector-based or heterologous vaccination regimens, respectively. The frequency of Spike-specific T-cell responses was comparably lower than CoVac-1-induced T-cell responses on day 28 after vaccination: median calculated spot count of 141 for mRNA-based vaccines, 24 spots for vector-based vaccines, and 98 spots for heterologous vaccination

regimens versus 488 spots for CoVac-1. This data was included in the new Extended Data Fig. 3c and the Results section of the revised manuscript (lines 165-168).

Comment 4: *T cell affinity measurement can be one other additional information that can help to better define the T cells induced by this novel peptide-based vaccine. I cannot find any indication in the material and methods of the concentration of peptides used to stimulate PBMC in the different assays. The authors should add this information.*

Author reply: We thank the reviewer for pointing out the missing information on peptide concentrations used to stimulate the PBMCs of study participants and human COVID-19 convalescents. To optimize readability, we included this information, which was formerly only stated in the Supplementary Appendix, in the Materials and Methods section of the revised main manuscript (lines 723-726 and 730-731).

Comment 5: *In addition, it will be of interest to show that the T cells induced by this peptide-vaccine are not low affinity T cells. An analysis of the functional affinity of the SARS-CoV-2 T cells induced by the peptide-based vaccine should be added. A simple experiment performed with different concentration of peptides in few selected vaccinated can indicate whether T cells recognize high or low quantity of peptides. A comparison with SARS-CoV-2 T cells induced by natural infection can also complete the characterization of the vaccine-primed T cells*

Author reply: We thank the reviewer for this valuable suggestion on how to strengthen our data and thus improve the manuscript. As suggested by the reviewer, we performed *ex vivo* IFN- γ ELISPOT assays with PBMCs from study participants (n = 5) collected on day 28 after vaccination and PBMCs from human convalescents (HCs, n = 5) collected on day 39-48 (median 44) after positive PCR and stimulated those with decreasing concentrations of our CoVac-1 peptides as well as for HCs with cross-reactive and SARS-CoV-2-specific T-cell epitope compositions, characterized in a previous study (Nelde *et al.* Nat Immunol, 2021). Titration with decreasing peptide concentrations (2.5 $\mu\text{g}/\text{mL}$ to 0.1 ng/mL) revealed detection of CoVac-1 peptides by vaccine-induced T cells down to 1 ng/mL (10 ng/mL 5/5 pCoVs, 1 ng/mL 3/5 pCoVs). This was even lower than the detection limits of SARS-CoV-2-specific T cells in HCs for CoVac-1 vaccine peptides (10 ng/mL 4/5 HCs, 1 ng/mL 0/5 HCs), SARS-CoV-2-specific (10 ng/mL 5/5 HCs, 1 ng/mL 0/5 HCs), and cross-reactive T-cell epitopes (10 ng/mL 2/5 HCs, 1 ng/mL 0/5 HCs). These insights are included in our revised manuscript (lines 158-164) as well as in the new Extended Data Fig. 3b.

We would like to thank the reviewer for the highly appreciated input on how to further improve our manuscript and hope that you are content with the additional experiments, data analyses, and revisions we made.

Referee #2:

Title: Phase I Trial of a Multi-Peptide COVID-19 Vaccine for the Induction of SARS-CoV2 T-cell Immunity

A. Summary of the key results

The manuscript by Heitmann et.al reports the results of phase I clinical trial aimed to evaluate safety, tolerability, and immunogenicity of CoVac-1, a novel multi-peptide COVID-19 vaccine, administered with the toll-like receptor (TLR)1/2 agonist XS15 emulsified in Montanide™ ISA51 VG to healthy adults ages 18-80 as a subcutaneous single injection. Overall, CoVac-1 vaccination demonstrated a favorable safety profile. Solicited local reactogenicity was mild to moderate in severity with several instances of severe erythema and swelling. All study participants developed granuloma at the injection site, which was reported as an expected outcome due to the use of the novel TLR1/2 XS15 adjuvant. Solicited systemic reactogenicity was absent or mild. CoVac-1 vaccine was shown to be highly immunogenic and induced polyfunctional CoVac-1 specific CD4+ and CD8+ T cells with high levels of expression of IFN- γ , TNF- α , and IL-2 28 days after vaccination. Notably, the authors claimed that SARS-CoV2 variants of concern (VOC) do not impact vaccine immunogenicity.

Author reply: We thank the reviewer for the kind review of our manuscript and highly appreciate the input on how to further improve our work. Please find below a detailed description on how we addressed the raised points.

Comment 1: *Originality and significance: if not novel, please include reference*

One of the main strengths of the manuscript is that it addresses the important and timely subject of designing a vaccine that can potentially induce rapid, broad-spectrum, and long-lasting immunity against COVID-19, even in populations with B cell deficiencies, both congenital and acquired, as well as the elderly. The authors designed an innovative peptide-based vaccine utilizing immunotherapy principles. Unlike many SARS-CoV2 vaccines that almost solely rely on SARS-CoV2 Spike glycoprotein (S) to induce immune response, CoVac-1 vaccine incorporates 6 dominant T cell epitopes derived from viral proteins that are associated with COVID-19 immunity. This is an interesting approach that could lead to advanced protection against the infection. This trial also incorporates the use of a novel, previously untested in clinical trials, adjuvant TLR1/2 ligand XS15, which was specifically designed to induced CD4+ and CD8+ immune responses. Furthermore, the results of this trial suggest that this vaccination strategy could potentially provide optimal protection, regardless of emerging VOC. However, the durability of immune responses was shown up to day 28 only, and therefore it is unclear whether this vaccine offers long term protection or if there is a chance of breakthrough infection.

Author reply: We fully agree that follow-up assessment of CoVac-1-induced T-cell responses is of utmost importance to prove vaccine-induced long-term T-cell immunity. Therefore, the study protocol

comprises follow-up sample collections and T-cell response analyses beyond day 28 after vaccination. To follow the reviewer's suggestion, which was likewise raised by Referee #1, we included follow-up data on vaccine-induced T-cell responses analyzed *ex vivo* and after 12-day *in vitro* expansion on day 56 as well as at month 3 post CoVac-1 vaccination in the revised manuscript (lines 147-152, Extended Data Fig. 3a). We could show that CoVac-1-induced T-cell responses persisted in the follow-up analyses at day 56 and month 3 after vaccination in all participants of Part I and Part II, with a decreasing IFN- γ T-cell response intensity observed *ex vivo* in Part I participants over time. However, profound and equivalent expandability of CoVac-1-induced T cells in both Part I and Part II participants was observed at month 3 compared to day 28 post vaccination, thereby indicating effective T-cell response upon virus challenge despite the expected decrease of *ex vivo* circulating CoVac-1-specific T cells over time (Dan *et al.*, Science, 2021; Bertoletti *et al.*, Cell Mol Immunol, 2021).

Comment 2: Data & methodology: validity of approach, quality of data, quality of presentation

In terms of trial design, dose rationale for either peptides or an adjuvant was not clearly described in the manuscript and therefore makes it difficult for the reader to understand why this trial was not a dose escalation study with several doses being evaluated or at what lowest dose the immunogenicity could be achieved. Authors are encouraged to revise the methodology section to provide clarity. No data or description of methodology or evaluation of the "sentinel" case was provided.

Author reply: We apologize for the lack of detailed information on trial design in terms of dose rationale for vaccine peptides and the adjuvant XS15. Doses of vaccine peptides were selected based on extensive experiences from former peptide-based vaccine trials including dose-finding studies. The dose of XS15, which was applied as an adjuvant for the first time in this clinical study, was based on preclinical *ex vivo* and *in vivo* analyses (including extensive murine toxicity data) as well as first clinical experiences in single volunteers (Rammensee *et al.*, JIC, 2019; Rammensee *et al.*, Vaccines, 2021). We added this information to the Materials and Methods section (lines 684-700) as well as to the Supplementary Appendix (page 7-9) of our revised manuscript. Furthermore, we included a more detailed description on the sentinel dosing of the first study participant in the Methods section of the revised manuscript (lines 635-639) and the Supplementary Study Results of the revised Supplementary Appendix (page 10).

Main Material and Method section:

Dose rationale

The dosage of CoVac-1 vaccine peptides was determined based on results from various clinical trials evaluating peptide vaccines¹⁻⁷ (including dose-finding studies for viral T-cell epitopes),

which showed significantly stronger immune responses to 250-500 µg versus 100 µg peptide dose, without significantly higher immune responses in the 1,000 µg versus 500 µg dose group³. Similar T-cell responses were induced with 250 µg and 500 µg peptide doses. Regarding safety, even doses up to 30 mg per peptide did not raise any concerns⁴. Based on these data, the dose of 250 µg per peptide was used for CoVac-1 vaccine peptides.

The dosage of the TLR1/2 agonist XS15 was determined based on *in vitro* analyses of immune cell activation by TLR1/2. In these assays, 10 µg/mL XS15 was shown to be the most efficient dose for the stimulation of immune cells. Considering the formation of a granuloma after subcutaneous injection of XS15 emulsified in Montanide™ ISA51 VG, which leads to a size-dependent decrease of XS15 concentration⁸, 50 µg XS15 were selected to achieve the desired dosage of 10 µg/mL at the vaccination site. In a toxicity study in mice, 50 µg XS15 in Montanide™ ISA51 VG, applied subcutaneously, did not reveal any local or systemic toxicity beyond the long known and expected local toxicity of Montanide^{9,10}. For a more detailed description of the dosage rationale for the vaccine peptides and the adjuvant please refer to the Supplementary Appendix.

Sentinel dosing

As safety measure, sentinel dosing of the first participant treated in Part I was conducted with a follow-up period of 28 days after vaccination followed by a sponsor safety assessment prior to proceeding with the vaccination of further study participants. Safety assessment of the sentinel dosing participant is described in detail in the Supplementary Appendix

Supplementary Appendix:

Dose rationale for vaccine peptides

Previous vaccination trials were performed at peptide doses ranging from 10 to 5,000 µg per peptide per vaccination. Even though only a few of these trials included a dose finding element, there is a tendency that doses below 100 µg are not effective to induce T-cell responses whilst doses above 500 µg do not seem to generate an increasing immunogenicity. Dose-finding studies performed with viral protein-derived epitopes showed significantly stronger immune responses in the 250-500 µg range versus the 100 µg dose, without significantly higher immune responses in the 1,000 µg vs. 500 µg group³. Concerning safety of peptide vaccines in different doses, no severe side effects were observed even with very high doses of peptides up to 30 mg^{4,5}. This is supported by own data of the investigator from various completed and ongoing trials (NCT02802943, NCT04688385, NCT0214922, NCT01265901)^{1,2}. Furthermore, a multi-peptide vaccination study for influenza evaluated safety and immunogenicity with two doses of

peptides (250 µg and 500 µg). No differences in the safety profile but a significant induction of functional T-cell responses was observed for both peptide dosages, suggesting 250 µg of peptide to be sufficient and safe for a prophylactic viral peptide vaccine⁶. This is further supported by first clinical data from a healthy volunteer vaccinated with viral peptide vaccines (240-300 µg per peptide) including two of the CoVac-1 peptides (250 µg) in combination with XS15 showing potent induction of peptide-specific T-cell responses and a good safety profile^{8,11}. Thus, the dose of 250 µg per peptide for the CoVac-1 vaccine was selected.

Dose rationale for XS15

The molecular mode of action of both the Pam₃Cys conjugates and XS15 is an activation of immune cells via the TLR1/2. These immune cells are mainly found in the blood and lymphoid tissues. Due to the XS15 and TLR1/2 interaction, desired as well as toxic effects are expected exclusively from these cells, in particular through an overactivation, which could then lead to a cytokine release syndrome. The dosage of XS15 is based on an in vitro assay that investigated both potential toxicity as well as efficiency. In this assay 10 µg/mL XS15 was shown to be most efficient for the stimulation of immune cells. The local formation of a granuloma (size up to 8 mL on day 17 after vaccination)⁸ locally after subcutaneous (s.c.) injection of XS15 emulsified in Montanide™ ISA51 VG, in a total volume of 500 µL suspension, leads to a size-dependent decrease of XS15 concentration. A dose of 50 µg XS15 was selected and achieved the desired concentration of 10 µg/mL at the vaccination site.

In a subsequent toxicity study in mice, a dose of 50 µg XS15 in Montanide™, applied locally s.c. did not reveal any local or systemic toxicity beyond the long known and expected local toxicity of Montanide™ alone. Furthermore, regarding systemic toxicity after s.c. injection of this XS15 dose the following considerations were made: in the absence of Montanide™ ISA51 VG and immediate distribution in the blood (6 L), a maximum blood concentration of 0.008 µg/mL would be expected. At this concentration, no measurable stimulation of immune cells is detected in the above-described in vitro test. When used with Montanide™, the formation of a granuloma at the injection site, which has a depot effect for peptides, a gradual release of these peptides or XS15 into the blood can be expected. Therefore, the actual blood concentration of XS15 after administration of 50 µg in a Montanide™/water emulsion is likely to be much lower than the maximum concentration of 0.008 µg/mL described above. Hence, a systemic toxic effect of XS15 is not expected at a dosage of 50 µg s.c. with or without Montanide™, which was proven in first clinical vaccination experiments in a healthy volunteer^{8,11}.

Dose rationale for Montanide ISA51 VG

Montanide™ ISA51 VG has been used in about 300 clinical trials from Phase I to Phase III, which represent >19,000 vaccine doses. Dosing of 0.5 mL after 50/50 mixture with peptides in

water is based on two published clinical studies evaluating influenza vaccines in > 2,500 donors showing high immunogenicity and a good safety profile

Safety assessment of sentinel dosing

Sentinel dosing took place at the end of November 2020. The safety assessment was performed on day 28 after vaccination. Until day 28 no SAE was reported. The participant developed erythema (grade 1), granuloma (grade 1), swelling (grade 1), itching (grade 1), and vaccination site lymphadenopathy (grade 1). Based on these observed AEs, the sponsor decided to continue recruiting.

Comment 3: Appropriate use of statistics and treatment of uncertainties

The statistical analysis section presented in the manuscript merits some significant revisions. As written (line 285-290, S141-146), it appears that the only statistical analysis performed was the sample size calculation; no subsequent statistical analyses of any data sets were described. It is very important for the authors to use appropriate statistical methodologies to provide a complete representation of the data acquired in the trial. This would include proper statistical comparisons/test between the cohorts and between different time points, description of how fold changes were calculated, and providing p-values where relevant. Main and extended figures need to be revised to include relevant statistical comparisons. Also, please consider evaluating the differences between % of CoVac-1 specific CD4 vs CD8 T cells, and the differences seen in specific cytokine production between the cohorts.

Author reply: As suggested by the reviewer we conducted statistical analyses of safety and immunogenicity data obtained in this trial. The respective results (p-values) are provided within the revised figures and the comparison of safety and immunogenicity data (including cytokine production of CoVac-1-specific T cells) between the study cohorts (Part I and Part II) is described within two new Extended Data Tables 1 and 3. Furthermore, we included the information on the used statistical tests in the revised Materials and Methods section and the respective figure and table legends. The data points used to calculate the indicated fold-changes were included in the revised Results section (lines 141-142, 177-183, 219-220).

Comment 4: Conclusions: robustness, validity, reliability

The authors seem to claim that CoVac-1 vaccination provides a “superior T cell immunity” (Line 190, 218), however this generalization is not fully supported by the data presented in the manuscript. Although, study results demonstrated that the vaccine induced polyfunctional T cell responses that are indeed correlates of protection against the disease, these responses are reported up to day 28 only. The authors also note the efficacy of the investigational candidate (line 74,269) but the study was not designed or powered to demonstrate efficacy. The authors are encouraged to revise the language to improve readability and avoid overstatements.

Author reply: We thank the reviewer for raising this point. We removed overstatements and claims of superiority in the revised manuscript. Furthermore, as suggested by the reviewer we replaced the misleading term efficacy by immunogenicity.

Comment 5: Suggested improvements: experiments, data for possible revision

Is it possible to develop antibodies against vaccine epitopes following immunization with CoVac-1? Was this evaluated? Could the authors comment on this?

Vaccination with CoVac-1 is intended to stimulate cellular immunity, however did the authors evaluate SARS-CoV2 specific antibody responses, were they induced by the vaccine?

Author reply: We thank the reviewer for raising this interesting issue and agree that, in theory, beside induction of T-cell immunity, also a humoral immune response in terms of antibodies directed against the CoVac-1 peptides might occur. We would, however, like to point out that our CoVac-1 peptides were deliberately selected either from non-surface proteins of the virus (and their subunits) or – in case of the spike-derived T-cell epitope P3_spi – from buried/hidden amino acid sequences, which based on the conformational state of the spike protein are not accessible to antibodies. Accordingly, in the particular case of our CoVac-1 vaccine, it is not conceivable that potentially induced antibodies would have functional anti-viral relevance.

When we evaluated, as suggested by the reviewer, whether CoVac-1 stimulates SARS-CoV-2-specific antibody responses, we observed that two participants of our Part II cohort/group had developed low titers of SARS-CoV-2 anti-spike IgG antibodies on day 28 after vaccination. Acute infection was excluded by negative results in sequential SARS-CoV-2 PCRs. We attribute this to the documented profound response of CD4⁺ T-cells induced by CoVac-1, which not only stimulate B cells to produce antibodies upon virus encounter but may also boost production of preexisting cross-reactive SARS-CoV-2 antibodies. Prevalence of the latter has been reported in 3-15% of unexposed individuals (Ng *et al.*, Science, 2020). Our respective results and a discussion of the relevance of potential antibodies against CoVac-1 vaccine peptides are now incorporated in the revised manuscript (lines 198-200, 258-265, 660-663), the Supplementary Appendix (page 7) and the new Extended Data Fig. 3d.

Comment 6: *Including immunological data beyond day 28 will strengthen the manuscript and will be a welcome addition if available.*

Author reply: As stated above we fully agree with the reviewer that a follow-up assessment of CoVac-1-induced T-cell responses is of utmost importance to prove vaccine induced long-term T-cell immunity. Therefore, we included follow-up data on vaccine induced T-cell responses analyzed on day

56 as well as at month 3 post CoVac-1 vaccination in the revised manuscript (Extended Data Fig. 3a). For more details, please refer to the reply to Comment 1.

Comment 7: *Line 79-87: Please, include percentage of males and females in the study as well as mean (SD) age*

Author reply: As suggested by the reviewer we included the information on the percentage of male and female participants in the study as well as their mean age with standard deviation in the revised manuscript (lines 95-98).

Comment 8: *Line 90: What was the duration of diary cards?*

Author reply: The diary cards were filled by all participants until day 28 after vaccination. We added this information in the revised manuscript (line 106).

Comment 9: *Line 93,95, 102: Was there any variation in reported reactogenicity between the age groups? How long did it take for grade 3 AE to resolve? Please, specify.*

Author reply: We thank the reviewer for raising these important questions. To identify any variation in the reported reactogenicity between Part I and Part II we conduct statistical analyses on the safety data. These data are summarized in a new Extended Data Table 1. We could show that reactogenicity did not differ between both study parts. This information was added to the Results section of the revised manuscript (lines 118-122).

Severe AEs (grade 3) comprised local erythema in 19% accompanied by severe swelling in 6% of all participants. These grade 3 AEs resolved within 2 days (median, range 1-7). This information was added to the Results section of the revised manuscript (lines 112-113).

Comment 10: *Lines 94-95, 97-98: How long did it take for granulomas/skin ulcerations to resolve?*

Author reply: The asymptomatic granulomas persisted throughout day 56 without affecting daily life activities of study subjects, showing a continuous decrease in size (up to 70% (median 30%) on day 56 calculated from the maximum size of the granuloma). These long-lasting granulomas represent a solicited and intended local reaction after Montanide-based vaccination (Aucouturier *et al.*, Expert Rev Vaccines, 2002; Lee *et al.*, J Clin Oncol, 2001; Van Doorn *et al.*, Hum Vaccin Immunother, 2016) enabling continuous local stimulation of SARS-CoV-2-specific T cells and thus induction of long-lasting T-cell responses without systemic inflammation.

Ulcerations, in terms of small skin defects occurred between day 28 and day 56 in reported cases and healed within 20 days (median, range 15-23) until day 56 not requiring any surgical intervention or

drug treatment. We specified the duration of these adverse events in the Results section of the revised manuscript (lines 115-118).

Comment 11: *Line 129: Briefly state rationale for a 12-day in vitro expansion protocol.*

Author reply: Additional 12-day *in vitro* expansion of peptide-specific T cells was performed to enable the detection and further characterization of low-frequent vaccine-induced and preexisting SARS-CoV-2-specific T cells, that were previously shown to be undetectable in *ex vivo* analyses (Nelde *et al.*, Nat Immunol.,2021; Lübke *et al.*, J Exp Med, 2020). Furthermore *in vitro* expansion was applied to prove the expandability of CoVac-1-induced T cells, which is of central importance for potent T-cell responses upon virus exposure (Swain *et al.*, Nat Rev Immunol, 2012; Strutt *et al.*, Nat Med, 2010). We stated this rationale for the 12-day *in vitro* expansion of peptide-specific T cells in the revised Materials and Methods section of the manuscript (lines 717-723).

Comment 12: *Line 141: Were there any vaccine induced CD8+ T cell response observed ex vivo or they were too low frequency to detect?*

Author reply: We thank the reviewer for making this point. Indeed, we observed *ex vivo* induction (defined as frequency of CoVac-1-specific T cells \geq 2-fold higher after vaccination compared to baseline prior to vaccination) of very low-frequent CoVac-1-specific CD8⁺ T-cell responses on day 28 after vaccination in 71% (5/7) and 69% (9/13) of analyzed participants of Part I and Part II, respectively. However, only 5% (1/20) of these T-cell responses passed our quality criterion for positivity (frequency of CoVac-1-specific T cells \geq 3-fold higher than the respective negative control). To enable a detailed characterization of the CoVac-1-induced CD8⁺ T cells we thus decided to expand the T cells *in vitro* before analysis. We included the data on *ex vivo* analyses of CoVac-1-induced CD8⁺ T cells in a new Supplementary Table S8 and described the rationale for characterization of CD8⁺ T cells after *in vitro* expansion in the revised Methods section of the manuscript (lines 720-723).

Comment 13: *Please, consider adding a supplemental table to complement Figure 1 that would include actual percentages of subjects that developed each of the local and systemic reactogenicity events.*

Author reply: We thank the reviewer for this valuable suggestion. We added the Extended Data Table 1 comprising the actual percentages of subjects that developed each of the local and systemic reactogenicity events to complement Figure 1 in the revised manuscript.

Comment 14: *Figure 2B: in the legend, please specify what method was used to measure CoVac-1 specific peptides*

Author reply: We apologize for the missing information in the figure legend. We clarified the methods used for every panel in Figure 2 in the revised figure legend: For Figure 2a-c we used IFN- γ ELISPOT assays, for Figure 2d intracellular cytokine and surface marker staining was used.

Comment 15: *Figure 2D: please revise y-axis label to clearly show that the cells were CoVac-1 specific CD4 T cells*

Author reply: We apologize for the unclear description of the depicted data and revised the y-axis label as suggested to “% of CoVac-1-specific CD4⁺ T cells” in Figure 2d as well as in the Extended Data Fig. 5 and 6c.

Comment 16: *Extended figure 4B demonstrates the data of 12-day in vitro expansion of CoVac-1 induced CD4 T cells, is there similar data available for human convalescent samples?*

Author reply: We thank the reviewer for making this point. The comparison of CoVac-1-induced T-cell responses with SARS-CoV-2-specific T-cell responses in human COVID-19 convalescents (HCs) was initially performed *ex vivo*. However, we absolutely agree with the reviewer that it is important to also compare expandability of vaccine-induced T-cell responses and SARS-CoV-2 T-cell responses after natural infection. In a previous work, IFN- γ ELISPOT data from HCs stimulated with SARS-CoV-2-specific and cross-reactive HLA class I and HLA-DR epitope compositions (Nelde *et al.*, Nature Immunol, 2021) showed a 12-fold (mean spot count 31 to 386) and 2-fold (190 to 349) expansion of peptide-specific T cells for HLA class I and a 9-fold (76 to 698) and 14-fold (69 to 988) expansion for HLA-DR after 12-day stimulation, respectively (Figure 1, for reviewer only).

Figure 1. Intensity of T cell responses.

T cell response intensity shown as mean calculated spot counts revealing cross-reactive as well as SARS-CoV-2-specific HLA class I and HLA-DR epitope compositions, directly *ex vivo* and after a 12-day *in vitro* expansion in samples of SARS-CoV-2 convalescent donors (n = 47). Each dot represents a single donor. Paired samples are connected by continuous lines, two-sided Wilcoxon test. Only donors with a detected T-cell response are depicted. T-cell responses were considered positive when mean spot counts were at least 3-fold higher than the respective negative control.

To further address this point and to provide comparable data of samples from HCs to data on CoVac-1-induced T-cell functionality in study participants shown in Extended Data Figure 5c we performed intracellular cytokine and surface marker staining for HCs (n = 9, sample collection 29-42 days after positive PCR, Supplementary Table S2) after 12-day *in vitro* expansion with CoVac-1 peptides. Alike in study participants, frequency of functional CD4⁺ T cells could be increased up to 105-fold after *in vitro* expansion (1.05% vs. 0.01% (median positive samples) CoVac-1-specific IFN- γ ⁺ CD4⁺ T cells). Of note, frequency of functional CD4⁺ T cells after *in vitro* expansion in study participants after CoVac-1 vaccination reached up to 15 times higher levels compared to HCs (18.6% vs. 1.23% (median positive samples) CoVac-1-specific TNF⁺CD4⁺ T cells, Part II participants vs. HCs, respectively). These data further underscore the potent expandability of CoVac-1-induced T cells after vaccination, which is of central importance upon SARS-CoV-2 exposure.

The novel data were included in the revised Extended Data Figure 5b and described in the results section of the revised manuscript (lines 179-183).

Comment 17: References: appropriate credit to previous work?

Yes, however, please, review journal specific guidelines to reduce number of references accordingly

Author reply: We thank the reviewer for making this point, we reduced the number of references to 50 for consistency with the journal's guidelines.

Comment 18: Clarity and context: lucidity of abstract/summary, appropriateness of abstract, introduction and conclusions

Major: The authors are recommended to revise the structuring of the paper to better guide the reader through introduction, methods, and results in advance of discussion. It is difficult to follow the logic of the trial design without being provided sufficient background information within the manuscript (e.g., the rationale for utilizing TLR ½ XS15 adjuvant and Montanide, selecting vaccine dose and regimen, peptide combination, CoVac-1 studies in pre-clinical models if available); while reading Line 59-63: the rationale for combining these 6 specific peptides does not seem immediately clear, please briefly discuss how recognizing multiple peptides could contribute to the development of a stronger immunity. Furthermore, is this the first report of XS15 adjuvant use in a human clinical trial? This also needs to be clarified.

Author reply: We thank the reviewer for this valuable suggestion on how to optimize the readability of the manuscript. Based on the Comment 2 we already included a detailed description of the dose rationale for CoVac-1 vaccine peptides, XS15, and MontanideTM ISA51 VG in the revised Materials and Method section (lines 684-700) as well as in the Supplementary Appendix (pages 7-9), including

information and references on pre-clinical data (for more details, please refer to the Author reply of Comment 2). We further added a description of the rationale for selecting the adjuvants XS15 and Montanide™ ISA51 VG in the revised Materials and Methods section (lines 668-680) as well as more detailed information on the selection of CoVac-1 peptides. Furthermore, we included a statement that this is the first report of the TLR1/2 ligand XS15 used in a clinical trial.

To further optimize the readability, we expanded the introduction of the manuscript (lines 62-76) to provide the reader sufficient background information on CoVac-1 peptide and adjuvant selection supported by pre-clinical data before presenting the results.

We hope that the reviewer is content with the revisions made. Please find below the novel sections in the revised Introduction and Material and Methods:

Introduction:

CoVac-1 is a peptide-based vaccine candidate designed to induce, upon one single vaccination, a broad and long-lasting SARS-CoV-2 T-cell immunity resembling that acquired by natural infection, and is not affected by evolving viral variants of concern (VOC). Thus, CoVac-1 is composed of multiple SARS-CoV-2 human leukocyte antigen (HLA)-DR T-cell epitopes derived from various viral proteins (spike, nucleocapsid, membrane, envelope, open reading frame (ORF) 8) that were proven in pre-clinical analyses to be (i) frequently and HLA-independently recognized by T cells in COVID-19 convalescents, (ii) of pathophysiological relevance for T-cell immunity to combat COVID-19, and (iii) to mediate long-term immunity after infection^{12,13}. Furthermore, CoVac-1 HLA-DR T-cell epitopes were selected to contain embedded HLA class I epitopes for induction of both, CD4⁺ and CD8⁺ T-cell responses. CoVac-1 vaccine peptides are adjuvanted with the novel toll-like receptor (TLR) 1/2 agonist XS15 emulsified in Montanide™ ISA51 VG, which showed in previous works to endorse activation and maturation of antigen presenting cells and prevent vaccine peptides from immediate degradation by forming a depot at the administration site, enabling the induction of an effective and potent T-cell response^{8,9,11}.

T cells play an important role for COVID-19 outcome and maintenance of long-term SARS-CoV-2 immunity, even in complete absence of humoral immune responses^{12,14-22}. In particular, the diversity of T-cell responses, i.e. the recognition of multiple T-cell epitopes, was shown to be of central importance for anti-viral defense and disease outcome in viral infections including SARS-CoV-2^{12,23-25}.

Material and Methods:

Trial vaccine and adjuvant

CoVac-1, developed and produced by the Good Manufacturing Practices (GMP) Peptide Laboratory of the Department of Immunology, University Tübingen, is a peptide-based vaccine comprising six HLA-DR-restricted SARS-CoV-2 peptides (Supplementary Table S1) derived from various SARS-CoV-2 proteins (spike, nucleocapsid, membrane, envelope, and ORF 8) and the adjuvant lipopeptide synthetic TLR1/2 ligand XS15⁸ (manufactured by Bachem AG, Bubendorf, Switzerland) emulsified in MontanideTM ISA51 VG⁹ (manufactured by Seppic, Paris, France). CoVac-1 peptides represent dominant SARS-CoV-2 T-cell epitopes (peptide-specific T-cell responses detected in > 50% and up to 100% of convalescents after SARS-CoV-2 infection) validated in human convalescents after SARS-CoV-2 infection to mediate long-term immunity^{12,13}. CoVac-1 peptides were predicted and validated to bind to multiple HLA-DR molecules (promiscuous binding)¹², which is important to enable HLA-independent induction of T-cell responses by CoVac-1^{12,13,26}.

CoVac-1 HLA-DR T-cell epitopes contain embedded HLA class I sequences for induction of both, CD4⁺ and CD8⁺ T-cell responses (Supplementary Table S1). CoVac-1 peptides were selected from viral non-surface proteins and their subunits or - in case of the spike protein-derived T-cell epitope P3_spi - from buried/hidden amino acid sequences, which are not accessible for antibodies in their conformational state. The linear 15-amino acid peptides are characterized by a free N-terminal amino group and a free C-terminal carboxy group. All amino acid residues are in the L-configuration and not chemically modified at any position. Synthetic peptides were manufactured by established solid phase peptide synthesis procedures using Fmoc chemistry^{1,27}.

The novel adjuvant XS15 hydrochloride is a water-soluble synthetic linear, 9-amino acid peptide with a palmitoylated N-terminus (Pam₃Cys-GDPKHPKSF)⁸. Acting as a TLR1/2 ligand, XS15 strongly activates antigen-presenting cells⁸ and enables the induction of strong *ex vivo* CD8⁺ and Th1 CD4⁺ responses to viral peptides, including SARS-CoV-2 T-cell epitopes, in preliminary *in vivo* analyses in a human volunteer upon a single subcutaneous injection of XS15 mixed to uncoupled viral peptides in a water-in-oil emulsion with MontanideTM ISA51 VG^{8,11}. This is the first report of the adjuvant XS15 being used in a human clinical trial. MontanideTM ISA51 VG is a mixture of a highly purified mineral oil (Drakeol 6VR) and a surfactant (Mannide monooleate). When mixed with an aqueous phase in a 50/50 ratio, it forms a water-in-oil emulsion. Such a Montanide-based water-in-oil emulsion has been used as vaccine adjuvant in multiple clinical trials^{9,10}, to build a depot at the vaccination site thereby preventing vaccine peptides from immediate degradation and thus enhancing the immune response.

CoVac-1 peptides (250 µg/peptide) and XS15 (50 µg) are prepared as a water-oil emulsion 1:1 with Montanide™ ISA51 VG to yield an injectable volume of 500 µL. Each participant received one subcutaneous injection of the CoVac-1 vaccine at the lower abdomen on day 1.

Comment 19: *Minor: Lines 38, 76, 81, 227: although the trial intended to enroll participants up to 80 years of age, the upper limit was 70 based on the data presented in Table 1. This needs to be noted somewhere in the manuscript.*

Author reply: Thank you for pointing out the misleading information on the participants' age. We added a statement on the maximum age of the participants in line 97-98 of the revised manuscript.

Comment 20: *Line 115, 120, 152: what methods were used here?*

Author reply: We apologize for the missing information and included a description of the methods used in the respective sections of the revised manuscript (lines 134-136, 212).

Comment 21: *Line 224: include clinicaltrials.gov identifier (NCT)*

Author reply: As suggested by the reviewer we included the clinicaltrials.gov identifier NCT04546841 in the revised manuscript (line 626).

Comment 22: *Line 245: please, specify the peptides used here*

Author reply: Thank you for pointing out this missing of information. The characteristics and sequences of the six CoVac-1 peptides are shown in Supplementary Table S1. We added a reference to Supplementary Table S1 in line 649 of the revised manuscript.

We would like to thank the reviewer for the highly appreciated input on how to further improve our manuscript and hope that you are content with the additional experiments, data analyses, and revisions we made.

References

- 1 Hilf, N. *et al.* Actively personalized vaccination trial for newly diagnosed glioblastoma. *Nature* **565**, 240–+, doi:10.1038/s41586-018-0810-y (2019).
- 2 Rini, B. I. *et al.* IMA901, a multi-peptide cancer vaccine, plus sunitinib versus sunitinib alone, as first-line therapy for advanced or metastatic renal cell carcinoma (IMPRINT): a multicentre, open-label, randomised, controlled, phase 3 trial. *Lancet Oncol* **17**, 1599-1611, doi:10.1016/S1470-2045(16)30408-9 (2016).
- 3 Kran, A. M. *et al.* HLA- and dose-dependent immunogenicity of a peptide-based HIV-1 immunotherapy candidate (Vacc-4x). *Aids* **18**, 1875-1883 (2004).
- 4 Sato, Y. *et al.* Immunological evaluation of peptide vaccination for patients with gastric cancer based on pre-existing cellular response to peptide. *Cancer science* **94**, 802-808 (2003).
- 5 Noguchi, M. *et al.* Induction of cellular and humoral immune responses to tumor cells and peptides in HLA-A24 positive hormone-refractory prostate cancer patients by peptide vaccination. *Prostate* **57**, 80-92, doi:10.1002/pros.10276 (2003).
- 6 Atsmon, J. *et al.* Safety and immunogenicity of multimeric-001--a novel universal influenza vaccine. *J Clin Immunol* **32**, 595-603, doi:10.1007/s10875-011-9632-5 (2012).
- 7 Feyerabend, S. *et al.* Novel multi-peptide vaccination in Hla-A2+ hormone sensitive patients with biochemical relapse of prostate cancer. *Prostate* **69**, 917-927, doi:10.1002/pros.20941 (2009).
- 8 Rammensee, H. G. *et al.* A new synthetic toll-like receptor 1/2 ligand is an efficient adjuvant for peptide vaccination in a human volunteer. *J Immunother Cancer* **7**, doi:10.1186/s40425-019-0796-5 (2019).
- 9 Aucouturier, J., Dupuis, L., Deville, S., Ascarateil, S. & Ganne, V. Montanide ISA 720 and 51: a new generation of water in oil emulsions as adjuvants for human vaccines. *Expert Rev Vaccines* **1**, 111-118, doi:10.1586/14760584.1.1.111 (2002).
- 10 van Doorn, E., Liu, H., Huckriede, A. & Hak, E. Safety and tolerability evaluation of the use of Montanide ISA51 as vaccine adjuvant: A systematic review. *Hum Vaccin Immunother* **12**, 159-169, doi:10.1080/21645515.2015.1071455 (2016).
- 11 Rammensee, H. G. *et al.* Designing a SARS-CoV-2 T-Cell-Inducing Vaccine for High-Risk Patient Groups. *Vaccines (Basel)* **9**, doi:10.3390/vaccines9050428 (2021).
- 12 Nelde, A. *et al.* SARS-CoV-2-derived peptides define heterologous and COVID-19-induced T cell recognition. *Nature immunology* **22**, 74-85 (2021).
- 13 Bilich, T. *et al.* T cell and antibody kinetics delineate SARS-CoV-2 peptides mediating long-term immune responses in COVID-19 convalescent individuals. *Sci Transl Med* **13**, doi:10.1126/scitranslmed.abf7517 (2021).
- 14 Rodda, L. B. *et al.* Functional SARS-CoV-2-Specific Immune Memory Persists after Mild COVID-19. *Cell* **184**, 169-183 e117, doi:10.1016/j.cell.2020.11.029 (2021).
- 15 Long, Q. X. *et al.* Clinical and immunological assessment of asymptomatic SARS-CoV-2 infections. *Nat Med* **26**, 1200-1204, doi:10.1038/s41591-020-0965-6 (2020).
- 16 Tan, A. T. *et al.* Early induction of functional SARS-CoV-2-specific T cells associates with rapid viral clearance and mild disease in COVID-19 patients. *Cell Rep* **34**, doi:10.1016/j.celrep.2021.108728 (2021).
- 17 Soresina, A. *et al.* Two X-linked agammaglobulinemia patients develop pneumonia as COVID-19 manifestation but recover. *Pediatr Allergy Immunol* **31**, 565-569, doi:10.1111/pai.13263 (2020).
- 18 Dan, J. M. *et al.* Immunological memory to SARS-CoV-2 assessed for up to 8 months after infection. *Science*, eabf4063, doi:10.1126/science.abf4063 (2021).
- 19 Le Bert, N. *et al.* SARS-CoV-2-specific T cell immunity in cases of COVID-19 and SARS, and uninfected controls. *Nature* **584**, 457-462 (2020).
- 20 Grifoni, A. *et al.* Targets of T cell responses to SARS-CoV-2 coronavirus in humans with COVID-19 disease and unexposed individuals. *Cell* **181**, 1489-1501. e1415 (2020).

- 21 Braun, J. *et al.* SARS-CoV-2-reactive T cells in healthy donors and patients with COVID-19. *Nature* **587**, 270-274 (2020).
- 22 Mateus, J. *et al.* Selective and cross-reactive SARS-CoV-2 T cell epitopes in unexposed humans. *Science* **370**, 89-94 (2020).
- 23 Messaoudi, I., Guevara Patino, J. A., Dyall, R., LeMaout, J. & Nikolich-Zugich, J. Direct link between mhc polymorphism, T cell avidity, and diversity in immune defense. *Science* **298**, 1797-1800, doi:10.1126/science.1076064 (2002).
- 24 Kiepiela, P. *et al.* CD8+ T-cell responses to different HIV proteins have discordant associations with viral load. *Nat Med* **13**, 46-53, doi:10.1038/nm1520 (2007).
- 25 Bilich, T. *et al.* Preexisting and post-COVID-19 immune responses to SARS-CoV-2 in cancer patients. *Cancer Discov*, doi:10.1158/2159-8290.CD-21-0191 (2021).
- 26 Tarke, A. *et al.* Comprehensive analysis of T cell immunodominance and immunoprevalence of SARS-CoV-2 epitopes in COVID-19 cases. *Cell Rep Med* **2**, 100204, doi:10.1016/j.xcrm.2021.100204 (2021).
- 27 Platten, M. *et al.* A vaccine targeting mutant IDH1 in newly diagnosed glioma. *Nature*, doi:10.1038/s41586-021-03363-z (2021).

Reviewer Reports on the First Revision:

Referee #1 (Remarks to the Author):

The authors provided a good quantity of data that fully answers my initial comments. The addition of data related to persistence of T cell response at 3 months, the comparison of T cell immunogenicity with other vaccines and the analysis of "T cell affinity" have in my opinion improved substantially the importance and the quantity of this work.

Referee #2 (Remarks to the Author):

Title Phase I Trial of a Multi-Peptide COVID-19 Vaccine for the Induction of SARS-CoV2 T-cell Immunity (Nature manuscript 2021-08-13293A).

The authors adequately addressed comments and concerns that we raised in our previous round of review and included new data in the revised manuscript that helped further support the conclusions of the study. To highlight, the authors demonstrated that the durability of CoVac-1-induced T cell responses lasted at least three months post vaccination. Furthermore, a comparison of the magnitude of T cell responses induced by peptide-based vaccine versus currently available SARS-CoV2 vaccines was performed which was a great addition to the manuscript. Revised statistical analysis section and appropriate statistical comparisons/tests were added to main and extended figures. The authors provided detailed and satisfactory answers to all our questions and improved